# BEGAN v3: Avoiding Mode Collapse in GANs Using Variational Inference

**Sung-Wook Park [1], Jun-Ho Huh [2,\*] and Jong-Chan Kim [1,\*]** 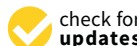

[1]  Department of Computer Engineering, Sunchon National University, 255 Jungang-ro, Suncheon-city, Jeollanam-do 57922, Korea; 411050@scnu.ac.kr

[2]  Department of Data Informatics, Korea Maritime and Ocean University, 727 Taejong-ro, Yeongdo-gu, Busan 49112, Korea

\*  Correspondence: 72networks@pukyong.ac.kr or 72networks@kmou.ac.kr (J.-H.H.); seaghost@sunchon.ac.kr (J.-C.K.); Tel.: +82-51-410-4347 (J.-H.H.); +82-61-750-3620 (J.-C.K.)

**Abstract:** In the field of deep learning, the generative model did not attract much attention until GANs (generative adversarial networks) appeared. In 2014, Google's Ian Goodfellow proposed a generative model called GANs. GANs use different structures and objective functions from the existing generative model. For example, GANs use two neural networks: a generator that creates a realistic image, and a discriminator that distinguishes whether the input is real or synthetic. If there are no problems in the training process, GANs can generate images that are difficult even for experts to distinguish in terms of authenticity. Currently, GANs are the most researched subject in the field of computer vision, which deals with the technology of image style translation, synthesis, and generation, and various models have been unveiled. The issues raised are also improving one by one. In image synthesis, BEGAN (Boundary Equilibrium Generative Adversarial Network), which outperforms the previously announced GANs, learns the latent space of the image, while balancing the generator and discriminator. Nonetheless, BEGAN also has a mode collapse wherein the generator generates only a few images or a single one. Although BEGAN-CS (Boundary Equilibrium Generative Adversarial Network with Constrained Space), which was improved in terms of loss function, was introduced, it did not solve the mode collapse. The discriminator structure of BEGAN-CS is AE (AutoEncoder), which cannot create a particularly useful or structured latent space. Compression performance is not good either. In this paper, this characteristic of AE is considered to be related to the occurrence of mode collapse. Thus, we used VAE (Variational AutoEncoder), which added statistical techniques to AE. As a result of the experiment, the proposed model did not cause mode collapse but converged to a better state than BEGAN-CS.

**Keywords:** deep learning; mode collapse; generative adversarial networks; boundary equilibrium generative adversarial networks; variational inference; computer vision; artificial intelligence

## 1. Introduction

The term deep learning has become so familiar [1]. Deep learning has rapidly expanded its range of use from AlphaGo's go match, which we all watched with interest, to professional jobs such as doctors and lawyers and to cultural and artistic fields that require creativity. There are countless cases of using deep learning, such as talking to AI voice secretaries on smart phones, receiving recommendations for the necessary products, preventing fraudulent credit card transactions, filtering spam mails, and detecting and diagnosing diseases. Global companies like Google, Facebook, Apple, Amazon, and IBM (International Business Machines) also invest heavily in researching and applying deep learning technology. Deep learning opened up new possibilities and became indispensable in everyday life.

The characteristics of deep learning models can be divided into discriminative and generative models for comparison. discriminative models distinguish and classify differences in input patterns. If one is to enter an image of a dog, the dog will be determined with a specific probability. The training model is trained to maximize the probability that a label called $Y$ will be output when given the data $X$. In other words, the discriminative model is a deep learning model that classifies or recognizes data and directly modeling conditional probability $p(y|x)$. Generative models, on the other hand, contain more information than discriminative models. Knowing the distribution of joint allows you to find the conditional probability and the distribution of the data itself. The generative model can understand and explain the structure of the input data.

Before the introduction of GANs (generative adversarial networks) in 2014, the generative model did not draw much attention in the deep learning field [2]. This is because loss calculation and the back-propagation learning of the generative model are difficult, and there was no methodology to raise the likelihood [3]. GANs use a structure and an objective function that are different from those of the previously introduced generative model. They use two neural networks called generator and discriminator. The generator generates images such as real, and the discriminator distinguishes whether they are real or synthetic. In the course of training, the generator learns more sophisticated synthetic techniques, and the discriminator grows into more accurate appraisers. GANs present a model that diligently refines the abilities of generators and discriminators.

The areas where GANs are utilized the most today are computer vision, such as image style, translation and image synthesis. Nowadays, they are also being used to generate non-image data such as voice and natural language [4,5]. The potential uses of GANs are growing.

GANs have introduced hundreds of applied models in recent years, with the problems cited improving. BEGAN (boundary equilibrium generative adversarial network), which performed better than the previously introduced GANs in image synthesis, learns the latent space of images while balancing and adjusting the generator and the discriminator [6].

In BEGAN, however, there is a fundamental problem with GANs called mode collapse. Although BEGAN-CS (Boundary Equilibrium Generative Adversarial Network with Constrained Space), which was improved in terms of loss function, was introduced, it did not solve the mode collapse [7].

Mode collapse is a phenomenon in which the generator generates only a few or a single image and is divided into partial collapse and complete collapse. In general, if the generator is updated every step, the discriminator initially assigns a low probability to the previous output of the generator, so the generator appears as a cycle of convergence or endless mode hopping. When a mode collapse occurs, the discriminator penalizes the images generated in the mode to increase the loss of the generator, and the generator moves to another mode to avoid it. This is called mode hopping. Mode collapse has emerged as a fundamental problem for GANs. Therefore, in this paper, research was conducted to alleviate or solve mode collapse.

Structural improvements in BEGAN-CS are also needed to address mode collapse. AE (AutoEncoder) cannot create particularly useful or well-structured latent space. Compression performance is not good either. In this paper, we saw these limitations in relation to the occurrence of mode collapse. Therefore, we used VAE (Variational AutoEncoder), which added statistical techniques to AE. The KLD (Kullback–Leibler Divergence) term was added to the loss function as a regularization loss that would create latent space well and reduce overfitting to the training data. In the KLD calculation process, BEGAN's hyperparameter $k$ was added. $k$ makes the decoded image equal to the original input, and it is included in the regularization loss calculation. In other words, the discriminator VAE is trained as two loss functions. We have also changed the structure of encoder and decoder. The activation function has been changed from ELU (Exponential Linear Unit) to Leaky ReLU (Leaky Rectified Linear Unit) [8,9]. Leaky ReLU, whose negative part is unsaturated, is thought to work better than ELU, whose negative part is saturated, so it was changed.

This paper presents an alternative research on the fundamental problem of GANs. Optimization studies were also conducted to improve the performance of BEGAN-CS, and problem solving was

sought. Finally, the GANs model with VAE as the discriminator was implemented to verify the performance of mode collapse, training instability, and evaluation criteria.

The implementation model was able to resolve mode collapse. In addition, we were able to learn a continuous, structural latent space to separate hair, hairstyle, hair color, skin, etc. Moreover, the blur phenomenon as a disadvantage of VAE did not occur.

The rest of this paper is organized as follows: Section 1 describes the background, purpose, and content and scope of the research; Section 2 presents the latest research related to the structure and application of GANs; Section 3 discusses the structure of BEGAN-CS, learning algorithms, and problems that arise during training; in Section 4, we compare and describe how the features of the proposed model designed for performance improvement are different from those of the existing models; Section 5 presents our experimentation and evaluation as to whether the performance of the proposed model improved as intended; Section 6 discusses the conclusion and future tasks.

## 2. Related Work

### 2.1. Structure of GANs

In the first announced GANs structure, both generator and discriminator consisted of fully connected layers. This type of structure can be trained with relatively simple datasets, such as the hand-written numeric dataset MNIST (Modified National Institute of Standards and Technology), CIFAR (Canadian Institute For Advanced Research)-10, and Toronto Face Dataset.

The convolutional GAN is a natural phenomenon because the CNN (Convolutional Neural Network) specializes in the field of computer vision. Training of the initial convolutional GAN using CIFAR-10 was more difficult than CNN. LAPGAN (Laplacian Pyramid of Generative Adversarial Network) decomposes the generation process on multiple scales and provides one solution [10]. In LAPGAN, the real data, which is the correct answer image, is itself decomposed into a Laplacian pyramid and is trained to generate layers from it.

Radford proposed DCGAN (Deep Convolutional Generative Adversarial Network) to improve the quality of the generated images [11].

DCGAN performs spatial down-sampling and up-sampling with stride convolution and transpose convolution. The operation above is useful when mapping from the image space to the low-dimensional latent space and discriminator because of the fast sampling rate and well-captured position change. Looking at the structure of Figure 1, the CNN and the operation process are reversed. For example, VGGNet receives an image measuring $3 \times 224 \times 224$ as input and outputs a vector of 1000 values, whereas DCGAN receives an image with 100 values as input and outputs an image measuring $3 \times 224 \times 224$ [12]. Such a difference is attributable to the fact that DCGAN is a generative model, whereas VGGNet is a discriminative model. Figure 1 shows the structure of DCGAN [11].

The generative model needs more understanding of the data generation process than the discriminative model [13]. MLP (multi-layer perceptron), deep MLP, and CNN are discriminative models. The discriminative model corresponds to supervised learning that can be learned only when feature vector $X = \{x_1, x_2, \ldots, x_n\}$ and label information $Y = \{y_1, y_2, \ldots, y_n\}$ are given as a training set. At this time, the learning algorithm does not need to find out the probability distribution of feature vector $x$. If we can only estimate the conditional probability $P(y|x)$, we can solve the classification or regression problem. Regression is an algorithm that models the relationship between one or more characteristics $x$ and successive target variables $y$. Characteristics are also called explanatory variables, and targets are response variables. On the other hand, the generation model estimates the probability distribution for vector $x$. This corresponds to unsupervised learning that does not require label information; if label information is available, it may be used. In summary, the discriminative model is supervised learning to estimate $P(y|x)$, and the generative model is unsupervised learning to estimate $P(x)$ or $P(x|y)$, $P(x, y)$.

Wu introduced GANs, which create 3D composite images, such as chairs, tables, and cars [14], and also suggested how to map the description or content of a 2D image to a 3D version.

Mirza extends the GANs framework to conditional settings by making the generator and discriminator class conditional [15]. Figure 2 shows the structure of cGAN (conditional generative adversarial nets).

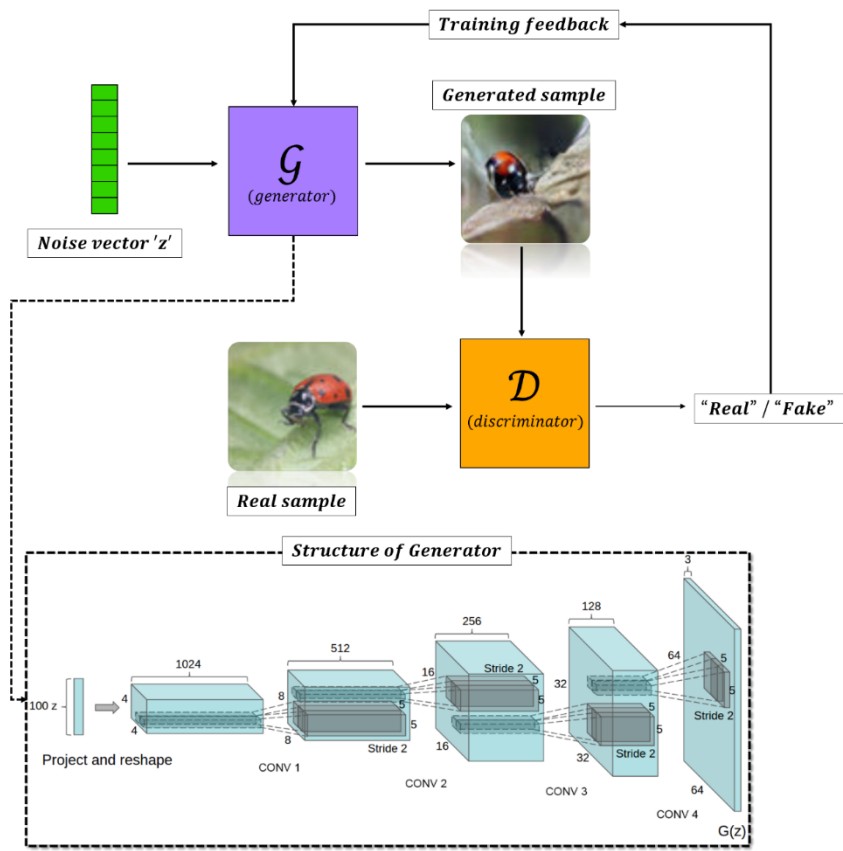

**Figure 1.** DCGAN (Deep Convolutional Generative Adversarial Network) generator used for LSUN (Large-Scale Scene Understanding) [11].

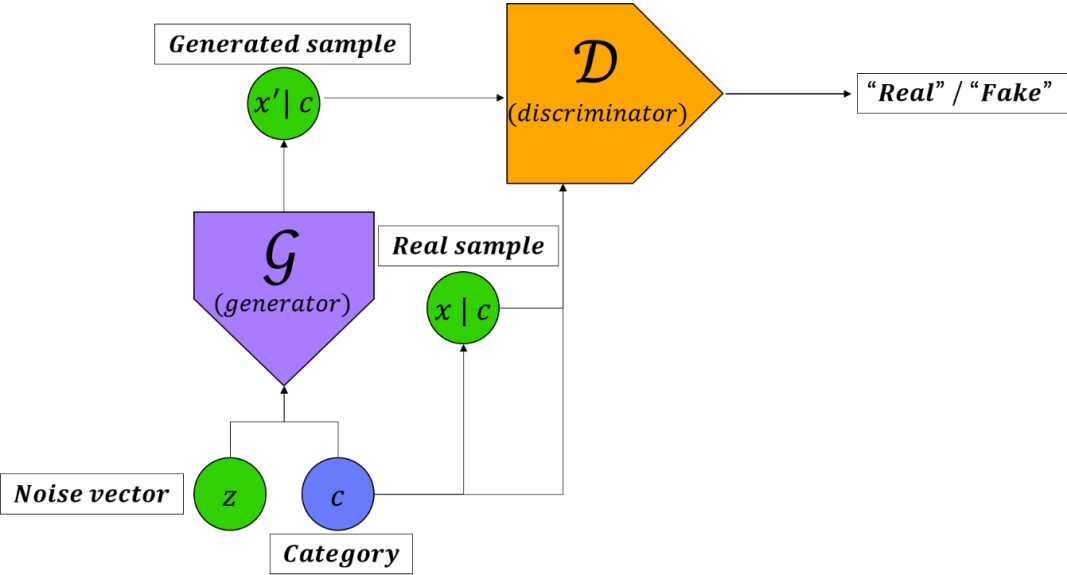

**Figure 2.** Conditional adversarial network.

The cGAN in Figure 2 performs the conditional distinguishing of real and synthetic images of discriminators. cGAN provide a better representation in a variety of data generation.

The discriminator in InfoGAN (Information maximizing Generative Adversarial Network) estimates the class label [16]. The expressions learned by InfoGAN are known to represent facial poses, lighting, and emotional changes well. The goal of the future research is to draw a parallel line between cGAN and InfoGAN to decompose the noise source into an Incompressible source and a latent code and to maximize the mutual information between the latent code and the generator output to discover the latent factor of the transformation. Latent code can be used to find object classes. Figure 3 shows the structure of InfoGAN.

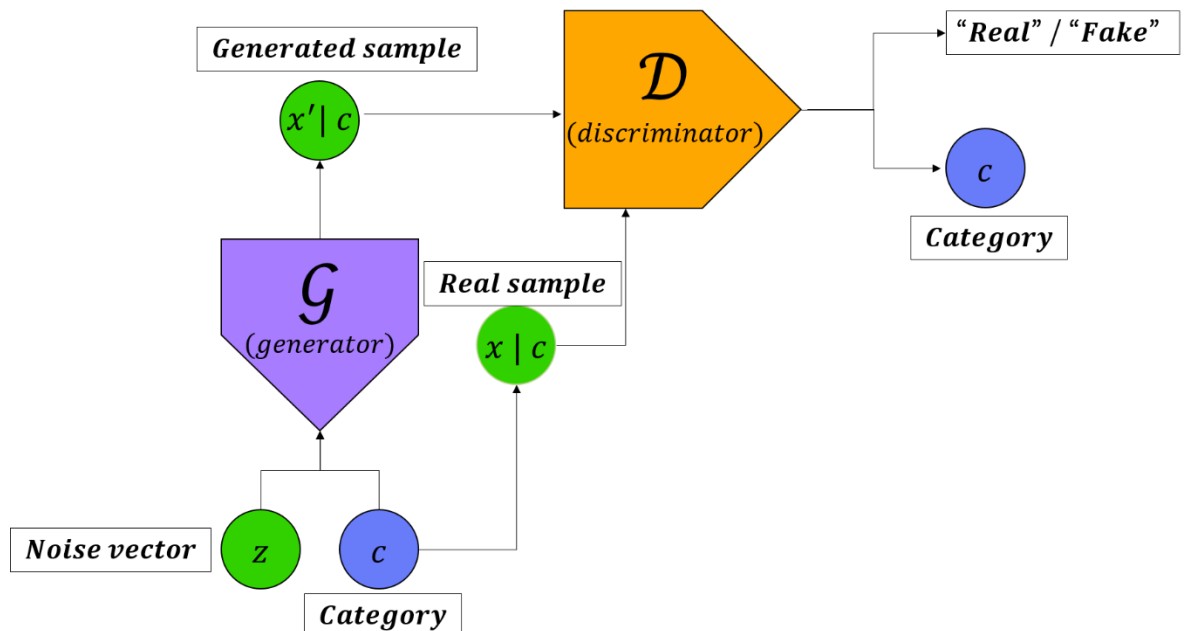

**Figure 3.** Structure and operation of InfoGAN (Information maximizing Generative Adversarial Network).

## 2.2. Application of GANs

Since GAN can quantitatively evaluate the features extracted from unsupervised learning, it can be applied to image classification. For example, if there is a constraint on the generated image, GAN can be applied to image synthesis. Constraints are conditions on how the training objectives should be achieved. Better super-resolution is possible by adding adversarial loss to the existing approach. It is also applicable to image-to-image, which automatically converts an input image into an output image. In other words, the application fields of GANs are very wide.

The trained GANs model can be used for other downstream tasks. Downstream is data transmitted from the upper medium to the lower medium. For example, the discriminator's convolution layer output can be used as a feature extractor and combined with a linear model like SVM (support vector machine). This is a structure wherein a feature vector that has passed through a feature extractor uses a classifier, such as SVM, as new input data. Radford achieved excellent classification performance when this method was applied to all supervised learning, unsupervised learning, and non-trained datasets [11].

Hostile training like ALI (Adversarially Learned Inference) can improve image quality when learning inference mechanisms simultaneously [17]. The representation vector generated in the last three hidden layers of the ALI encoder achieves a lower misclassification ratio than DCGAN. Higher performance was achieved when label information was added to the ALI.

When there is less labeled training data, GANs can be used to generate more training data. Shrivastava improved the synthetic image while maintaining annotation information [18] and

achieved state-of-the-art performance in posture and gaze estimation work with synthetic images only. Spatiotemporal GAN also reported good results for gaze estimation and prediction [19]. When a model trained as a synthetic image is applied to a real image, however, it does not always show good results [20].

Bousmalis proposed a method of matching the synthetic image of the source domain with the target domain [20]. Liu proposed a method of using multiple GANs with combined weights to synthesize images from different domains [21].

A significant part of the recent GANs research is to improve the quality and usefulness of the generated images. LAPGAN introduces cascades of convolutional networks to generate images in a rough way. Once initiated, Cascade refers to a series of stages wherein each stage is triggered due to the previous stage and the results are continued until the end. LAPGAN expanded the cGAN wherein the generator and the discriminator take additional label information as input. The idea was later extended to address the problem of natural language processing. Huang changed the algorithm above to work in intermediate expressions rather than low-resolution images [22].

Reed used GANs to synthesize the images with text description [23]. For example, if a text description such as "head is black, wing is orange, beak is a white bird" is the input of the network, GANs generate a plausible image.

For ground truth rows in Figure 4, the first entry corresponds, and is directly connected, to the caption; the next two entries are sampled from the same species.

In GAWWN (Generative Adversarial What-Where Network), the location of the image is determined according to the conditions [24]. GAWWN supports an interactive user interface that enables gradually drawing large images with a description of the object and a bounding box.

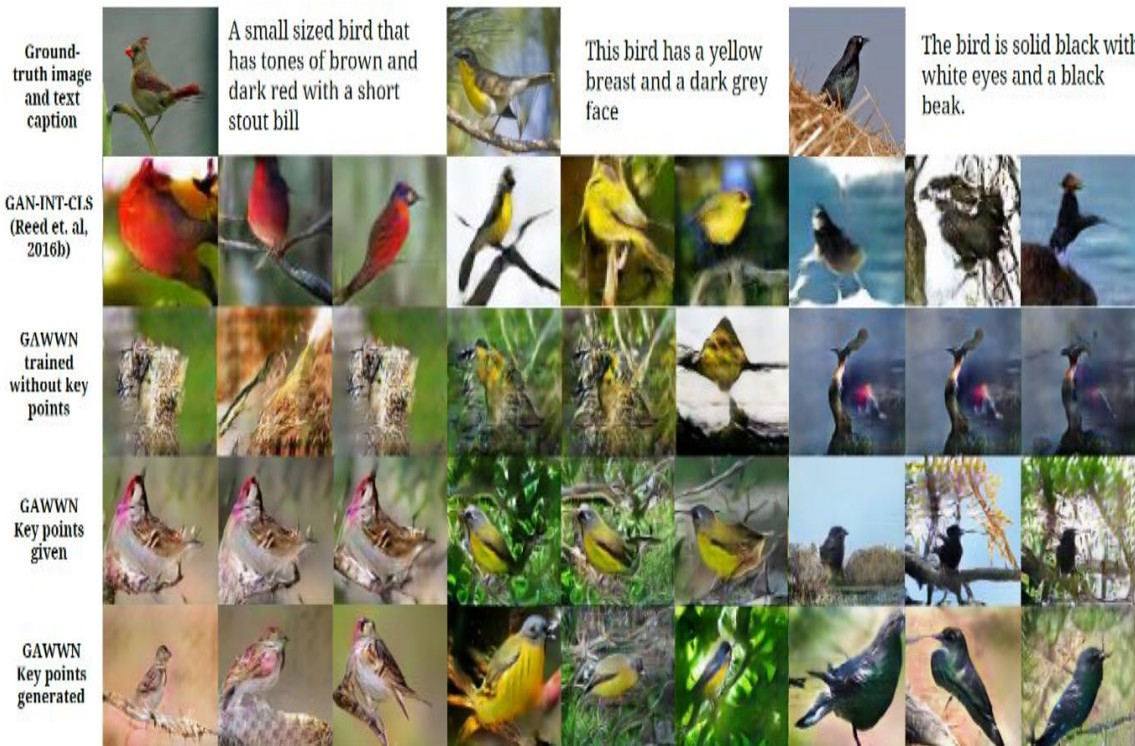

**Figure 4.** Comparison of GAWWN (Generative Adversarial What-Where Network) with GAN-INT-CLS from Reed, including the ground-truth images. GAN-INT is Learning with manifold interpolation, GAN-CLS is Matching-aware discriminator, and GAN-INT-CLS is a combination of GAN-INT and GAN-CLS [25].

This way, cGAN can synthesize new images with specific properties and develop tools that can intuitively edit images such as changing hairstyles, wearing glasses, and reducing age [25,26].

The Pix2Pix model is a model that maps from an input image to an output image [27]. Therefore, the training images are also a pair of inputs and outputs. This model showed excellent results in a variety of computer vision problems, such as semantic segmentation, map generation from aerial photographs, and the colorization of black and white images.

Wang suggested the idea of synthesizing the surface normal map as shown in Figure 5 and subsequently mapping the image to a natural scene.

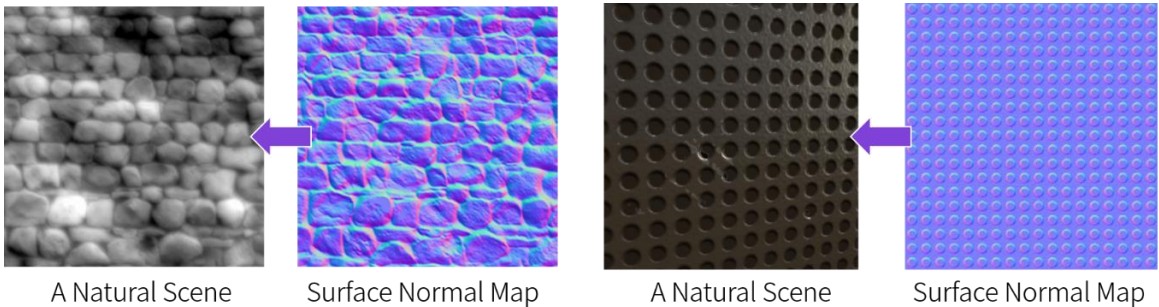

**Figure 5.** Surface normal map-to-natural scene mapping example.

CycleGAN (cycle-consistent generative adversarial network) introduced cycle-consistent loss to preserve the original image in the process of transformation and inverse transformation [28]. CycleGAN's training does not require matching pairs of images, unlike Pix2Pix. This solved the researchers' dilemma regarding large-scale data collection.

The artistic style transfer in Figure 6 trains on paintings and natural images and renders with painter-style images like Picasso or Monet [28]. Rendering is the process of converting two-dimensional or three-dimensional data described by numbers and equations into human-recognizable images.

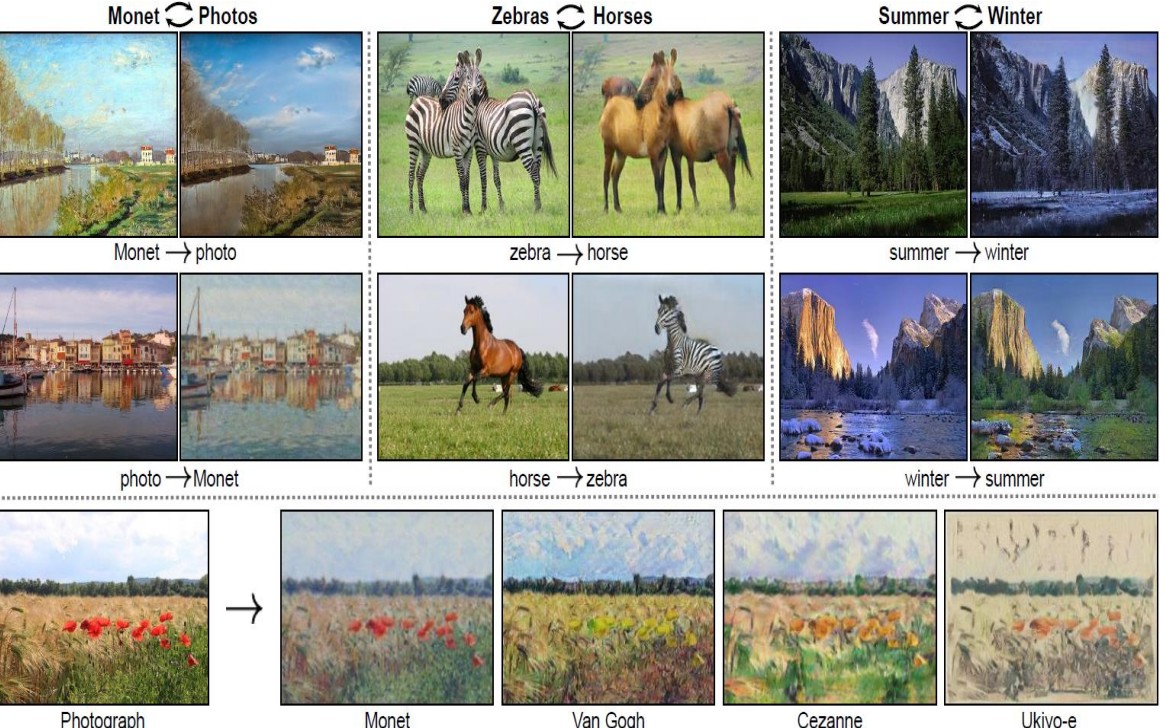

**Figure 6.** Artistic style transfer using cycle-consistent adversarial networks [28].

## 3. Boundary Equilibrium Generative Adversarial Network with Constrained Space

Both BEGAN and BEGAN-CS have not solved the mode collapse, but the proposed method is based on boundary equilibrium and constrained space algorithms. The characteristics of the proposed model are covered in the next chapter, and this chapter describes the structure and learning algorithm of BEGAN and BEGAN-CS.

BEGAN calculated the Wasserstein distance lower bound between the real data loss distribution in Equations (1) and (2) and the generated data loss distribution and studied the effect when the loss distribution was matched. Wasserstein distance is the minimum cost of moving A's probability distribution to B's probability distribution. Distribution is a function of probability density.

$$(v) = |v - D(v)|^{\eta} \text{ where } \begin{cases} D : R^{N_x} \mapsto R^{N_x} \\ \\ \eta \in 1, 2 \\ \\ v \in R^{N_x} \end{cases} \tag{1}$$

$\mathcal{L}$ means loss $R^{N_x} \mapsto R^{N_x}$ for training AE per pixel. $D$ denotes an AE function, $\eta$ is a target norm, and $v$ denotes $N_x$-dimensional image.

$$\mathcal{W}_1(\mu_1, \mu_2) = \inf_{\gamma \in \Gamma(\mu_1, \mu_2)} \mathbb{E}_{(x_1, x_2) \sim \gamma} [|x_1 - x_2|] \tag{2}$$

$\mathcal{W}$ is Wasserstein distance, $\mu_1, \mu_2$ is the distribution of AE losses, $\gamma$ is an element of $\Gamma$, and $\Gamma$ is the set of two distributions. inf is short for Infimum, the greatest lower bound. In other words, it is the largest value of the lower limit. $\mathbb{E}$ is the expected value as the average of the values that can be obtained by infinitely repeating a random process. ~ is a tilde, a symbol describing the relationship between variables and distribution. Distributions $\mu_1$ of $x_1$ and $\mu_2$ of $x_2$ are used to explain that it is drawn from $\gamma$.

$$\text{Inf } \mathbb{E}[|x_1 - x_2|] \geq \text{ Inf } \mathbb{E}[|x_1 - x_2|] = |m_1 - m_2| \tag{3}$$

Equation (3) is the Wasserstein distance, optimizing the lower bound of the Wasserstein distance between AE loss distributions. $m_1 - m_2 \in R$ is the means of representation. The Jensen inequality can be used to derive the lower bound of $\mathcal{W}_1(\mu_1, \mu_2)$.

The discriminator is designed to maximize the value calculated in Equation (1), which can be maximized through selection, or in Equation (4).

$$(a) \begin{cases} \mathcal{W}_1(\mu_1, \mu_2) \geq m_1 - m_2 \\ \\ m_1 \to \infty \\ \\ m_2 \to 0 \end{cases} \text{ or } (b) \begin{cases} \mathcal{W}_1(\mu_1, \mu_2) \geq m_2 - m_1 \\ \\ m_1 \to 0 \\ \\ m_2 \to \infty \end{cases} \tag{4}$$

$\mu_1$ is the loss distribution of $\mathcal{L}(x)$, $x$ is the real image, $\mu_2$ is the loss distribution of $\mathcal{L}(G(z))$, $G$ is a generating function with condition $R^{N_x} \mapsto R^+$, and $z \in [-1, 1]^{N_z}$ is the $N_z$-dimensional uniform distribution image. In general, we choose method $(b)$ because minimizing $m_1$ will naturally encode the real image.

Equation (5) expresses the problem as a purpose of GANs. Objective means the object to be optimized.

$$\begin{cases} \mathcal{L}_D = L(x; \theta_D) - L(G(z_D; \theta_G); \theta_D) & \text{for } \theta_D \\ \\ \mathcal{L}_G = -\mathcal{L}_D & \text{for } \theta_G \end{cases} \tag{5}$$

$\theta_D, \theta_G$ is the weight of the discriminator and generator and is updated to minimize loss $\mathcal{L}_D, \mathcal{L}_G$. $z_D, z_G$ is the output from $z$. $z$ is the noise vector, which is the input of the generator or discriminator. A later paper used abbreviated $G(\cdot) = G(\cdot, \theta_G)$ and $\mathcal{L}(\cdot) = \mathcal{L}(\cdot, \theta_D)$.

In general, the generator and the discriminator are not well-matched in terms of balance, with the discriminator winning easily. The concept of equilibrium is introduced to address this situation.

When the generator and the discriminator of GANs are in equilibrium, this can be expressed as Equation (6).

$$\mathbb{E}[\mathcal{L}(x)] = \mathbb{E}[\mathcal{L}(G(z))] \tag{6}$$

In Equation (6), the loss distribution when the discriminator cannot distinguish between the generated image and the real image should be the same as that when the predictive loss is included to equalize the performance of the generator and the discriminator. BEGAN attempted to balance the two models by introducing a new hyperparameter $\gamma$, as shown in Equation (7). $\gamma$ is a real number between 0 and 1.

$$\gamma = \frac{\mathbb{E}[\mathcal{L}(G(z))]}{\mathbb{E}[\mathcal{L}(x)]} \tag{7}$$

$\gamma$ is used as a variable to determine the diversity ratio. The lower values of $\gamma$ reduce the variety of images because the focus is on the automatic encoding of the real images.

As a result, the objective function of BEGAN becomes equal to Equation (8). The objective function means that the target function needs to be optimized.

$$\begin{cases} \mathcal{L}_D = \mathcal{L}(x) - k_t \cdot \mathcal{L}(G_{(Z_D)}) & \textit{for } \theta_D \\ \mathcal{L}_D = \mathcal{L}(G(z_G)) & \textit{for } \theta_G \\ k_{t+1} = k_t - \lambda_k(\gamma \mathcal{L}(x) - \mathcal{L}(G(z_G))) & \textit{for each training step 't'} \end{cases} \tag{8}$$

Equation (8) maintains equilibrium $\mathbb{E}[\mathcal{L}(G(z))] = \gamma \mathbb{E}[\mathcal{L}(x)]$ based on the theory of proportional control. The proportional control theory is one of the automatic control methods; the more the force deviates from the target point, the greater the force to return to the target point. Then, use $k_t$ to control how much emphasis is placed on $\mathcal{L}(G(z_D))$ as the gradient is falling. The range of variable $k_t$ is a real number between 0 and 1. At the start of training, $k_t$ is initialized to zero. To establish Equation (7), $k_t$ is used as a form of feedback control adjusted at each training stage. $\lambda_k$ is the learning rate for $k$.

The data generated during the initial training phase is close to zero; since the real data distribution has not been trained yet, it produces data that can be easily reconstructed in AE. This is denoted as $\mathcal{L}(x) > \mathcal{L}(G(z))$, and equilibrium may be constrained in subsequent training.

The approximation of Equation (3) and $\gamma$ of Equation (7) influences the Wasserstein distance modeling. Therefore, it is important to verify the data generated from various $\gamma$ values. $\theta_D, \theta_G$ is updated based on each loss.

The problem of mode collapse often occurs in BEGAN, so BEGAN-CS has been proposed to solve this problem as a model that adds the concept of latent space constraint as a loss function.

The images generated in the model share a latent vector similar to the real image. BEGAN-CS proposes latent space constraint loss $\mathcal{L}(c)$ using this feature, where $c$ is a constraint and loss limits the norm of the difference between latent vector $z$ and encoder $Enc(G(z))$.

Constrained losses during training are optimized only for the discriminator. Mode collapse occurs on the generator side but does not add constraint loss to the generator. Constraint loss is a regularizer that regards function $Enc(G(\cdot))$ as an identity function. This way, the encoder maintains the diversity and balance of randomly extracted $z \in Z$.

As objective functions, Equations (9) to (10) are similar to BEGAN except for the additional constraint loss.

$$\begin{cases} \mathcal{L}_G = L(G(z_G; \theta_G); \theta_D), & \textit{for } \theta_D \\ \\ \mathcal{L}_D = L(x_{real}; \theta_D) - k_t \cdot L(G(z_D; \theta_G); \theta_D) + \alpha \cdot \mathcal{L}_c, & \textit{for } \theta_G \end{cases} \tag{9}$$



$$\begin{cases} \mathcal{L}_c =\| z_D - Enc(G(z_D)) \|, & \textit{the constraint loss} \\ k_{t+1} = k_t + \lambda(\gamma \mathcal{L}(x; \theta_D) - \mathcal{L}(G(z_G; \theta_G); \theta_D)), & \textit{for the epoch} \end{cases} \tag{10}$$

Total loss $\mathcal{L}_c$ of the generator and total loss $\mathcal{L}_D$ of the discriminator are optimized to solve $\theta_D$ and $\theta_G$, respectively. The $\theta_D$-related function $\mathcal{L}(x; \theta_D) = \| x - D(x) \|$ calculates the norm of the difference between given image $x$ and reconstructed image $D(x)$ in the discriminator decoder. Latent vectors $z_D$ and $z_G$ are randomly generated from $z$. Variable $k_t \in [0, 1]$ controls the importance of $\mathcal{L}(G(z_D; \theta_G); \theta_D)$. Hyperparameter $\gamma$ maintains balance between loss $\mathcal{L}(x; \theta_D)$ of the real image and loss $\mathcal{L}(G(z_G; \theta_G); \theta_D)$ of the generated image. Hyperparameter $\alpha$ is the weight of constraint loss. Constraint loss $\mathcal{L}_c$ forces $Enc(G(\cdot))$ to be an identity function for $z_D$.

## 4. Proposed Avoiding Method

The weights of the proposed model are trained with two loss functions. Reconstruction losses make the decoded image the same as the original input, and regularization losses form the latent space well and reduce overfitting to the training data. The reconstruction loss uses L2 loss rather than cross-entropy, and KLD is used for regularization loss [29]. KLD is a function used in the information theory.

As a branch of applied mathematics that has been established to transmit discrete data to a noisy communication channel, the information theory provides a method of calculating the average length of message samples extracted from code optimization and specific probability distributions. Code optimization is a process applied to intermediate codes or object codes in order to minimize the wasted resources of the compiler. In deep learning, the information theory can be used to quantify the similarity of two probability distributions with a particular continuous probability variable that is difficult to interpret.

KLD calculates the difference between the real data distribution and the model-estimated data distribution. KLD does not have negative values. When probability distributions $P$ and $Q$ are present for two random variables, KLD is equal to Equation (11).

$$D_{KL}(P \parallel Q) = \sum_x P(x) \log \frac{P(x)}{Q(x)} = \left( -\sum_x P(x) \log Q(x) \right) - \left( -\sum_x P(x) \log P(x) \right) \tag{11}$$

The KLD of Equation (11) can be expressed as Equation (12) using expected value $\mathbb{E}$.

$$\sum_x P(x) \log \frac{P(x)}{Q(x)} = \mathbb{E}_{x \sim P}[\log P(x)] - \mathbb{E}_{x \sim P}[\log Q(x)] \tag{12}$$

The second term in Equation (12), $-\mathbb{E}_{x \sim P}[\log Q(x)]$, is cross-entropy. When the distribution of input data is $p_{data}$, and the distribution of data estimated by the model is $p_{model}$, Equation (12) can be derived as in Equation (13).

$$D_{KL}(p_{data} \parallel p_{model}) = \mathbb{E}_{x \sim p_{data}}[\log p_{data}(x) - \log p_{model}(x)] \tag{13}$$

$p_{data}$ in Equation (13) is a fixed value during training. Since Equation (13) aims to minimize the value, $p_{model}$ should be as similar as possible to $p_{data}$.

Variational inference [30] is a problem of finding an estimation distribution $q(z)$ that is close to the posterior distribution and can be expressed as in Equation (14).

$$q(z) = argmin_q D_{KL}(q(z) \parallel p(z|x)) \tag{14}$$

In Equation (14), KLD can derive the lower bound for marginal likelihood $p(x)$, as in Equation (15).

$$
\begin{aligned}
D_{KL}(q(z) \parallel p(z|x)) &= \mathbb{E}_{q(z)}[\log q(z) - \log p(z|x)] \\
&= \mathbb{E}_{q(z)}[\log q(z)] - \mathbb{E}_{q(z)}[\log p(z|x)] = \mathbb{E}_{q(z)}[\log q(z)] - \mathbb{E}_{q(z)}[\log p(z,x)] + \log p(x)
\end{aligned}
\tag{15}
$$

Equation (15) can be restructured into Equation (16).

$$
\begin{aligned}
\log p(x) &= D_{KL}(q(z) \parallel p(z|x)) - \mathbb{E}_{q(z)}[\log q(z)] + \mathbb{E}_{q(z)}[\log q(z,x)] \\
&\geq -\mathbb{E}_{q(z)}[\log q(z)] + \mathbb{E}_{q(z)}[\log p(z,x)] \\
&= -\mathbb{E}_{q(z)}[\log q(z)] + \mathbb{E}_{q(z)}[\log p(z)] + \mathbb{E}_{q(z)}[\log p(x|z)] \\
&= -D_{KL}(q(z) \parallel p(z)) + \mathbb{E}_{q(z)}[\log p(x|z)]
\end{aligned}
\tag{16}
$$

In Bayes' theorem, $p(x)$ is called evidence. Therefore, the lower bound of $\log p(x)$ in Equation (16) is called ELBO (Evidence Lower Bound).

The decoder of VAE is optimized as the $\log p(x)$ value increases. Nonetheless, optimization is not easy because there are many latent vectors ($z$). Variable inference can derive the lower bound of $\log p(x)$, as shown in Equation (17).

$$
\log p(x) \geq -D_{KL}(q(z|x) \parallel p(z)) + \mathbb{E}_{q(z)}[\log p(x|z)]
\tag{17}
$$

Variable inference assumes a single normal distribution for all data. The model is simple, but learning is difficult when data complexity is high. To solve this problem, VAE uses the parameter of $q$ as a function of input data $x$. Adjusting $q$ to maximize ELBO $-D_{KL}(q(z|x) \parallel p(z)) + \mathbb{E}_{q(z)}[\log p(x|z)]$ in Equation (17) changes the distribution of $q$ as input data $x$ changes. When using a gradient descent-based optimizer, the objective function of VAE becomes equal to Equation (18).

$$
\mathcal{L} = -\mathbb{E}_{q(z)}[\log p(x|z)] + D_{KL}(q(z|x) \parallel p(z))
\tag{18}
$$

The first term in Equation (18) is the reconstruction loss. The encoder receives input data $x$ and outputs $z$ from $q$. The decoder receives $z$ and restores $x\prime$. In other words, the reconstruction loss is the cross-entropy of input data $x$ and reconstruction data $x\prime$. The second term in Equation (18) is the regularization loss. Give prior $x$ the controllability of the output $z$ of the encoder. As such, the condition that $q(z|x)$ should be similar to $p(z)$. The first term induces convergence in the direction that the objective function is smaller with a larger value. The second term induces convergence such that the smaller the value is the smaller the objective function.

As the regularization loss of Equation (18), KLD can be broken down into Equation (19).

$$
\mathcal{L} = -\mathbb{E}_{q(z)}[\log p(x|z)] - H(q(z|x)) + H(q(z|x), p(z))
\tag{19}
$$

The second term in Equation (19) is the entropy of the posterior distribution. $H$ means entropy. The larger the diversity of A sampled in the distribution, the smaller the objective function is. The larger the diversity of $z$ sampled in the corresponding distribution, the smaller the objective function is. In addition, mode collapse is alleviated, and learning is stable. In Equation (19), the third term is the cross-entropy of prior and post distributions; the more similar the information, the smaller the objective function becomes.

We have changed the discriminator structure from AE to VAE, taking into account the stable training environment of VAE and the benefits of mitigating mode collapse. Because VAE uses two losses, there are variations in terms of loss function. If KLD is $D_{KL}(P(x) \parallel Q)$ when the real image is inputted to the encoder and KLD is $D_{KL}(P(G(z) \parallel Q)$ when the synthetic image is inputted to the encoder, the final regularization loss is equal to Equation (20).

$$
\mathcal{L}_R = D_{KL}(P(x) \parallel Q) - k \cdot D_{KL}(P(G(z) \parallel Q)
\tag{20}
$$

In the process of calculating $\mathcal{L}_R$ in Equation (20), BEGAN's hyperparameter $k$ was added since we determined that $k$, which determines the importance of real and synthetic images, is necessary for regularization loss. Subscript $R$ means regularization.

Reconstruction losses used L2 losses rather than cross-entropy. As a result, the objective function $\mathcal{L}_D$ of Equation (9) is changed to Equation (21).

$$\mathcal{L}_D = \frac{(\mathcal{L}(x_{real}; \theta_D) - k_t \cdot \mathcal{L}(G(z_D; \theta_G))) + \mathcal{L}_R}{2} + \alpha \cdot \mathcal{L}_c, \quad for \; \theta_G \tag{21}$$

We used VAE as the discriminator and matched the loss distribution, not the data distribution. Generator and discriminators train to balance. The proposed model is similar to the training objective of WGAN (Wasserstein GAN) [31]. It does not require the discriminator to be a K-Lipschitz sheet because it does not use the Kantorovich and Rubinstein duality theorem, as in BEGAN [32]. Figure 7 shows the discriminator structure of the proposed model.

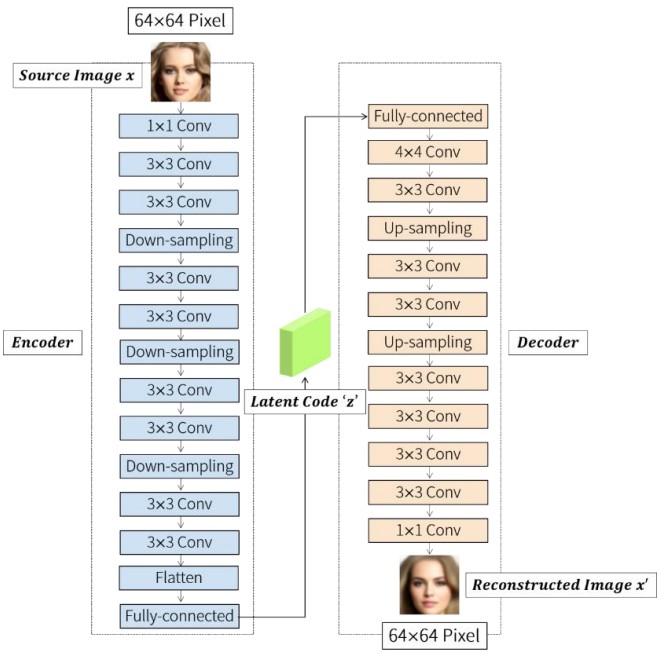

**Figure 7.** Discriminator structure of the proposed model.

The proposed model transforms the input image into a parameter of a specific statistical distribution, unlike BEGAN-CS, which compresses the input image with fixed coding of latent space. This assumes that the input image is generated through a statistical process. It also adds randomness to the encoding and decoding process. The proposed model extracts a random image from the normal distribution using mean $\mu$ and variance $\sigma$. This image is decoded and restored to the original input.

Randomness allows stable training and the encoding of meaningful representations anywhere in latent space. The encoder converts the input image into two parameters of the latent space: mean $\mu$ and variance $\sigma$. $z$ is randomly extracted from the normal distribution of the latent space assuming that the input image is generated. Because the encoder may output negative numbers, it is trained to output the log value of the variance rather than the standard deviation. The decoder maps $z$ of the latent space as the original input image and restores it. In the process of sampling $z$, epsilon parameters are used. This epsilon is made randomly. Therefore, points near the average are decoded into an image similar to the input image. This process makes the latent space a continuous, meaningful space. The latent space created by AE may be structural or non-continuous. The latent space created in VAE is structural and continuous, making it suitable for handling as concept vector. Figure 8 shows the principle of a variable autoencoder.

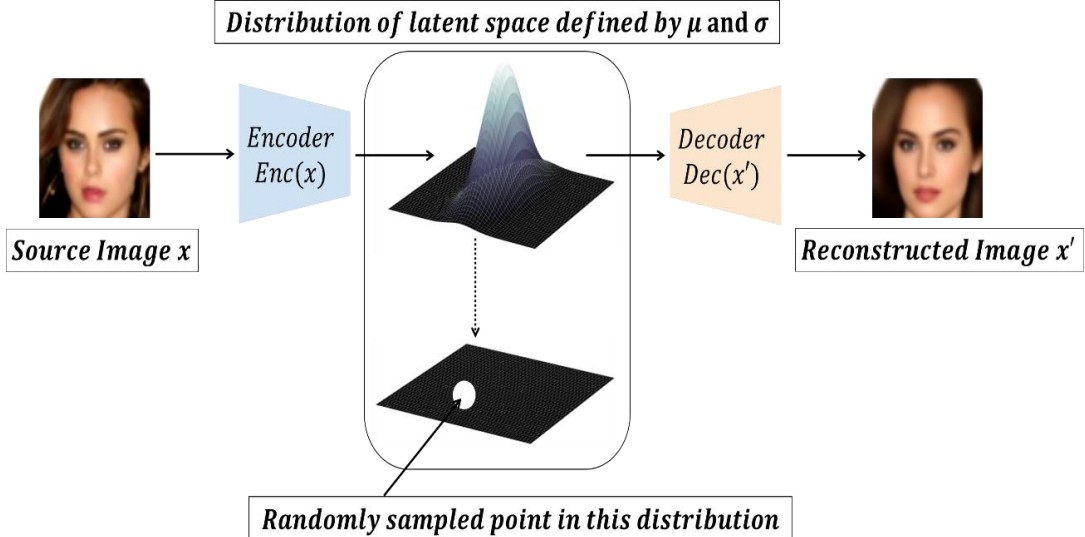

**Figure 8.** VAE (Variational AutoEncoder).

The convolution operation was used from 1 to 4, and the activation function was used in Leaky ReLU. "1" means $1 \times 1$ convolution, and "4" means $4 \times 4$ convolution. Since ReLU does not adjust the weight when the function is not active, the gradient is zero. When the gradient becomes zero and sparse, it can interfere with GANs training and slow down learning.

Sparse is often a desirable phenomenon in deep learning, but not GANs. Leaky ReLUs allow for negative active values, so sparse is alleviated. Considering such advantage, we changed the activation function from ELU to Leaky ReLU.

The Leaky ReLU function is defined as in Equation (22) and is shown in Figure 9 (right).

$$a_{i,j,k} = \max\!\left(z_{i,j,k},\, 0\right) + \lambda \min\!\left(z_{i,j,k},\, 0\right) \tag{22}$$

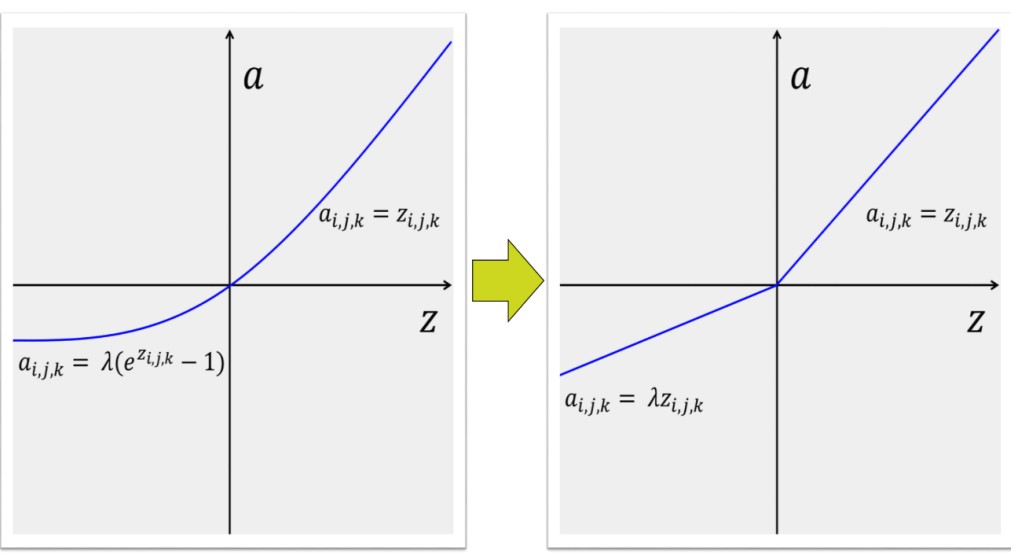

**Figure 9.** Exponential linear unit (**left**), leaky rectified linear unit (**right**).

In Equation (22), $\lambda$ is a hyperparameter with range of 0~1. Since Leaky ReLU compresses the negative part without mapping them to constant 0 like ReLU, a positive value less than 1 is outputted even if the function is inactive. In the experiment, the $\lambda$ value of Leaky ReLU was set to 0.2.

The ELU function is defined as in Equation (23) and is shown in Figure 9 (left).

$$a_{i,j,k} = \max\left(z_{i,j,k},\, 0\right) + \min(\lambda(e^{z_{i,j,k}} - 1), 0) \tag{23}$$

Leaky ReLU, PReLU (Parametric Rectified Linear Unit), and RLReLU (Randomized Leaky Rectified Linear Unit) denote the negative parts of the function graph in the unsaturated form, whereas ELU appears in the saturated form. In Equation (23), $a_{i,j,k}$ is the final output, $z_{i,j,k}$ is the input from the $k$ channel to the position $(i,\ j)$ active function, and $\lambda$ is a hyperparameter defined to control the saturation of the function for negative values. Since the performance of activation functions varies depending on the situation, it is difficult to determine the exact dominance relation between activation functions. Therefore, it is important to apply it according to the situation [33,34].

The number of filters in the convolution layer increases linearly with each down-sampling, which is done with stride 2, and up-sampling is performed by the $k$-NN (nearest neighbors) algorithm. Data processed at the encoder and decoder boundaries is mapped through the fully connected layer. Input vector $z$ is randomly extracted from the even distribution between $[-1, 1]$.

Adam (Adaptive Moment estimation) was used as the optimizer [35]. As an algorithm that adds RMSprop (Root Mean Square propagation) to momentum [36], Adam, like AdaGrad (Adaptive Gradient), accumulates the previous gradient but follows the exponential moving average of RMSProp [37]. Adam adjusts the update direction to increase the weight of the current gradient. This addresses the problem of AdaGrad. Adam can adaptively adjust the learning rate, and the weighted search path is more efficient than the stochastic gradient descent [38].

## 5. Experiment Result

The dataset used was CelebA (large-scale celeb faces' attributes), not the 360K Celebrity Face [39]. The CelebA dataset displays a person's identity with five facial signs and 40 attributes. There are 10,177 unique people with 202,599 facial images in the dataset. As one of the largest datasets available for face identification, detection, marking, and attribute recognition problems, CelebA is a good candidate for experimental datasets because it has many artifacts, such as aliasing, compression, and blur, which are difficult for the generator in terms of generation. In computer graphics, aliasing is a phenomenon wherein lines, etc., are limited due to resolution limitations. The mini batch size was set to 128, the epochs to 300, and the resolution of the training and prediction images to $64 \times 64$ pixels.

The optimizer used Adam and set learning rate = "0.0001", coefficient for primary momentum $\beta 1$ = "0.9", coefficient for secondary momentum $\beta 2$ = "0.999", and epsilon = $10^{-8}$. The bias was set to all zeros. The weight initialization used Glorot's uniform distribution. The weights were adjusted to constant per layer as the training progressed. The learning rate of the generator and the discriminator is the same.

The magnitude of input vector $z$ was set to 64, and $\gamma$ to 0.5. $\gamma$ was used as a variable to determine the diversity ratio. Weight $k_t$, which determines how important $\mathcal{L}(x)$ and $\mathcal{L}(G(z_D))$ are, was set to 0.0, and the learning rate $\lambda_k$ of $k_t$ was set to 0.001. $\alpha$, which determines the importance of the constraint loss, was set to 0.1. The remaining hyperparameters that were not mentioned are the same as those of BEGAN-CS. $x$ is the real image, and $\mathcal{L}(x)$ is the error for the real image. $z_D$ is a random vector to input into the generator when the discriminator learns, and $\mathcal{L}(G(z_D))$ is the error of the image generated by the generator.

In this experiment, we avoided the visual artifact problem by lowering the initial learning rate. The last layer of the encoder was kept at $8 \times 8$ size. $N_h$ and $N_z$ were set to 128 and 64, respectively, in the experiment. The model used in the experiment has 10,331,144 parameters. The training time was 8 days, 7 h, and 5 min when L2 loss was used and 8 days, 6 h, and 30 min when L1 loss was used.

For hardware specification, the CPU used was Intel Core i7 7700K Kaby Lake; the graphics card was NVIDIA TITAN Xp 12GB, the RAM was Samsung DDR4 48GB, and the SSD (Solid State Drive) was Samsung 850 Pro 512GB. Table 1 shows the hardware specifications for the experiment.

**Table 1.** Hardware specifications.

| Hardware | Specifications |
|----------|----------------|
| CPU | Intel Core i7 7700K |
| Graphics Card | NVIDIA TITAN Xp 12 GB |
| RAM | Samsung DDR4 48 GB |
| SSD | Samsung 850 Pro 512 GB |

For software specification, the operating system was ubuntu 16.04.4 LTS; the CUDA (Compute Unified Device Architecture) was 9.0.176, the cuDNN (cuda Deep Neural Network library) was 7.1, tensorflow was 1.12.0, and python was 3.5.2 [40]. Tensorflow is a framework for machine learning and deep learning. Table 2 presents the software specifications for the experiment.

**Table 2.** Software specifications.

| Software | Specifications |
|----------|----------------|
| Operating System | Ubuntu Linux 16.04.4 LTS |
| Programming Language | Python 3.5.2 |
| GPGPU | CUDA 9.0.176 |
| Deep Neural Network Library | cuDNN 7.1 |
| Deep Learning Framework | Tensorflow 1.12.0 |

Figures 10–15 are representative images of EBGAN (Energy-Based Generative Adversarial Network), CEGAN (Calibrating Energy-based Generative Adversarial Network), AVB (Adversarial Variational Bayes), and VEEGAN (Variational Encoder Enhancement to Generative Adversarial Network) as well as the proposed model [41–44]. Each image in Figures 10–15 is identical at 64 × 64 resolution. EBGAN is a model wherein the discriminator is an autoencoder, and CEGAN, AVB, and VEEGAN are models to which the variational principle is applied.

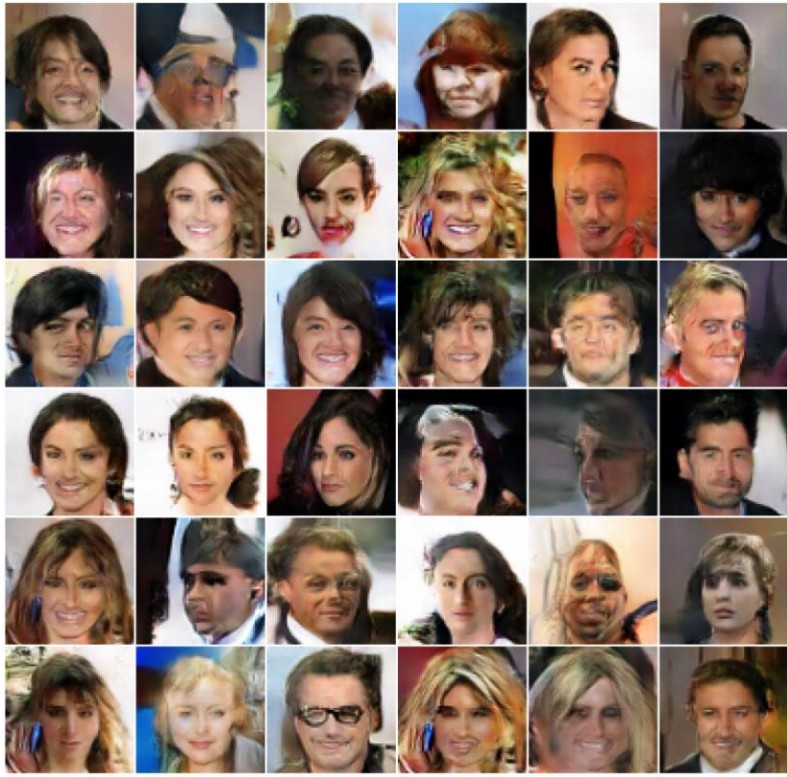

**Figure 10.** Representative images of EBGAN (Energy-Based Generative Adversarial Network) [41].

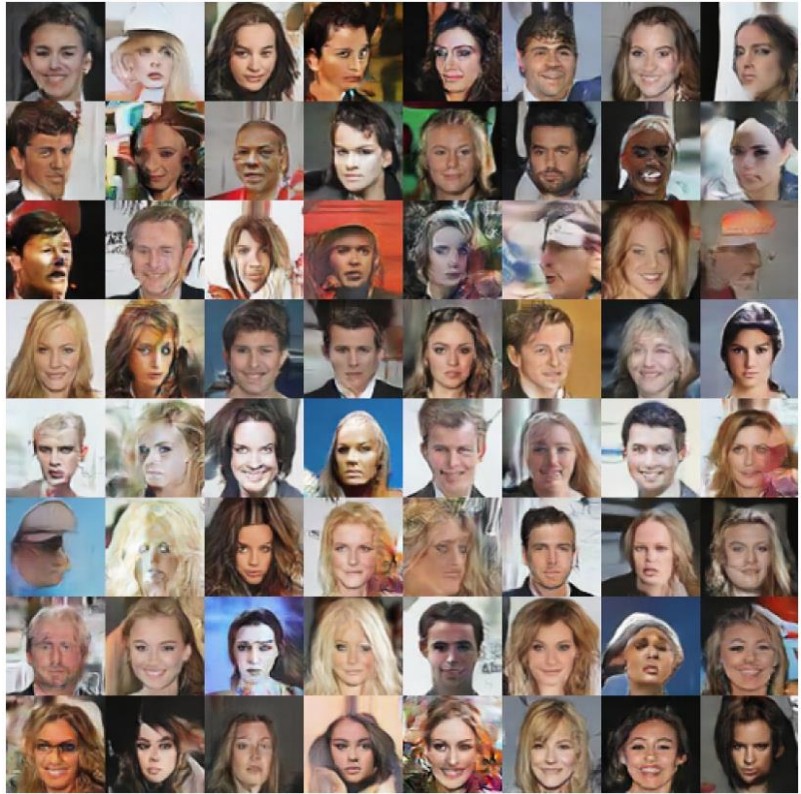

**Figure 11.** Representative images of CEGAN (Calibrating Energy-based Generative Adversarial Network) [42].

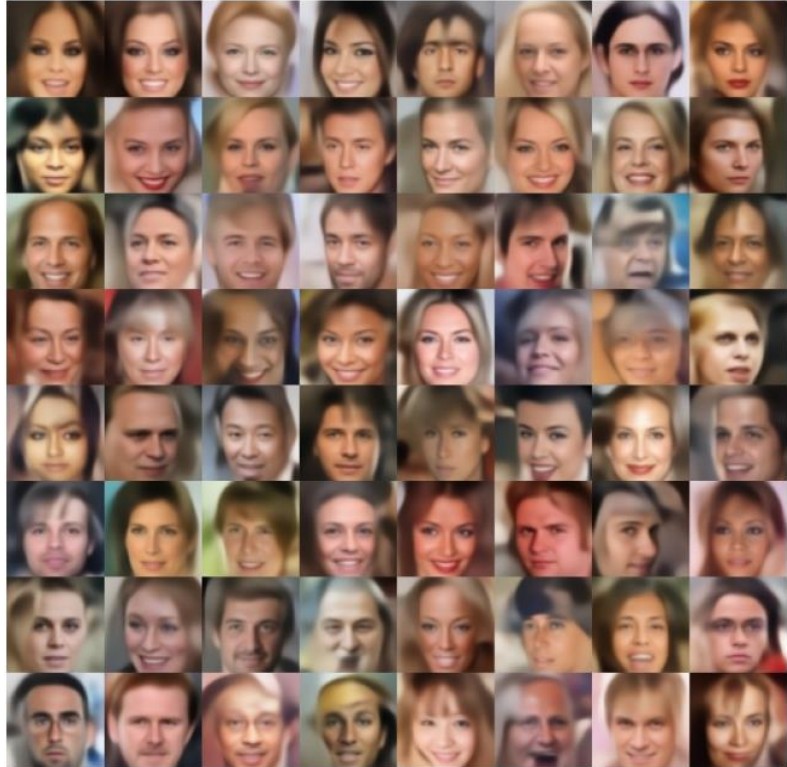

**Figure 12.** Representative images of AVB (Adversarial Variational Bayes) [43].

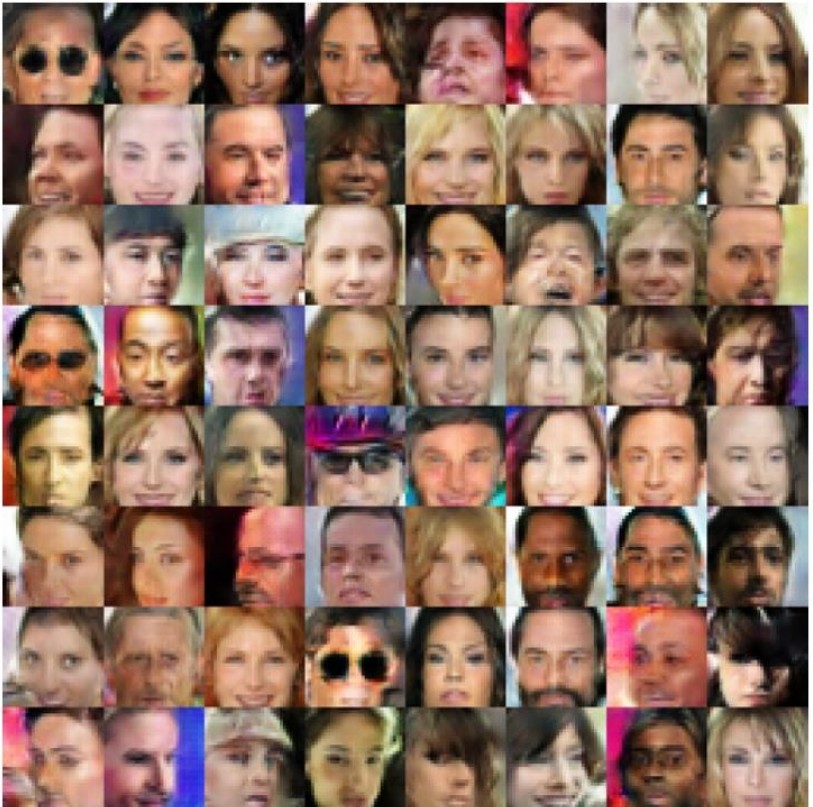

**Figure 13.** Representative images of VEEGAN (Variational Encoder Enhancement to Generative Adversarial Network) [44].

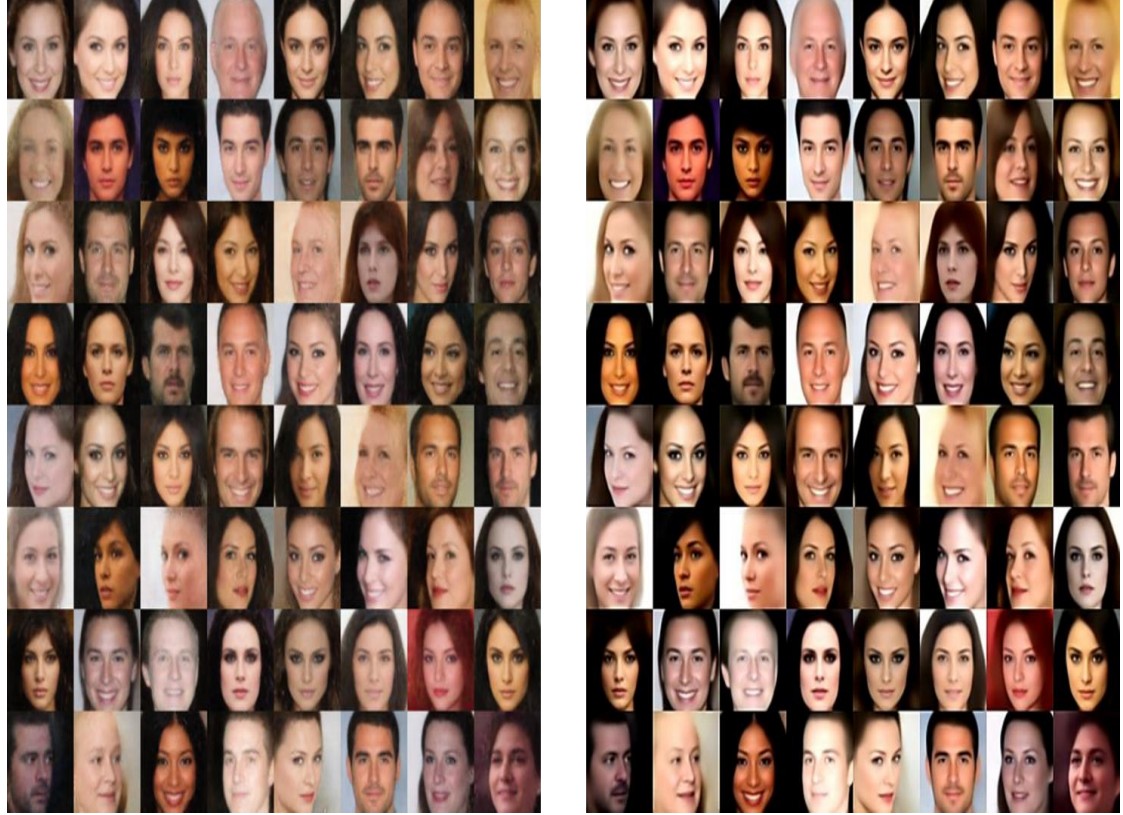

**Figure 14.** Representative images of *G* (**left**) and *VAE* (**right**) in the proposed model (L2 loss).

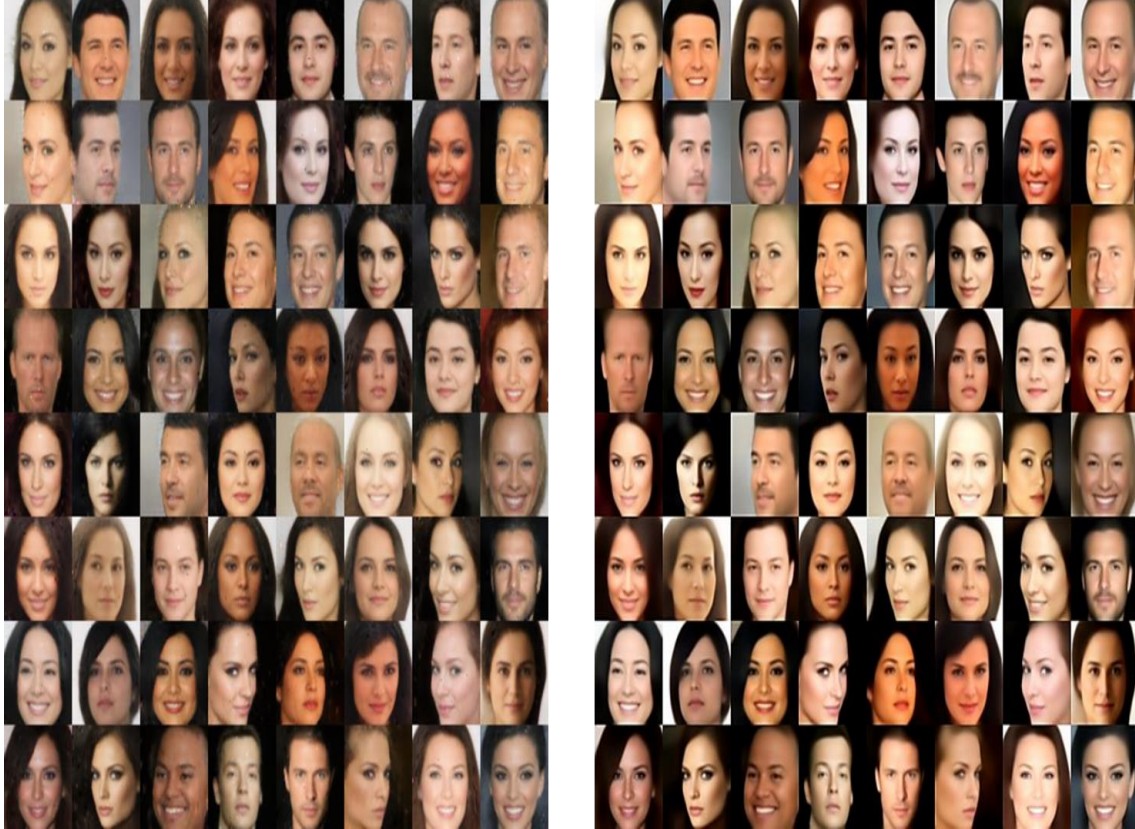

**Figure 15.** Representative images of *G* (**left**) and *VAE* (**right**) in the proposed model (L1 loss).

Figures 14 and 15 show the result of outputting 64 images of 64 × 64 pixels at a time. Unlike the representative images of other models, various face poses, facial expressions, genders, skin colors, and hairstyles can be observed in Figures 14 and 15, which present the images generated in 300 epochs. The experiments show that the combination of variational inference and equilibrium is superior to the comparative model. All models used CelebA as a training dataset. BEGAN-CS did not insert a representative image separately because mode collapse occurred.

In this study, the implementation model was verified for mode collapse, training instability, and evaluation criteria. As an evaluation criterion, we used $M_{global}$, which is designed to measure the convergence of the BEGAN model.

In BEGAN-CS, IS (Inception Score) and FID (Frechet Inception Distance) were used [45,46]. IS may misrepresent performance when generating only one image per class. In other words, the score may be high even if the images' diversity is low. FID cannot take into account precision, recall, and F1 score in the harmonic mean of precision and recall. For this reason, we did not use the two evaluation criteria in this study.

Figure 16 presents the convergence measurement results of BEGAN-CS. In Figure 16a, BEGAN-CS is most optimal when $M_{global}$ is 1.5. Nonetheless, it is difficult to determine whether the model converges or collapses with $M_{global}$ values. As can be seen from Section A, no evidence of mode collapse can be found from the convergence measure value.

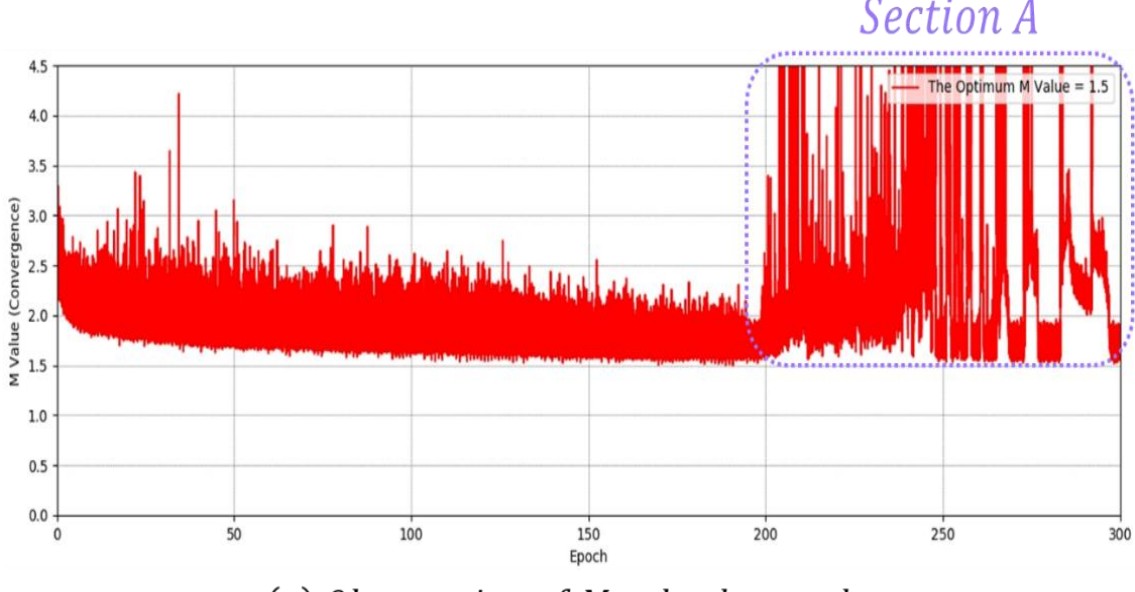

(a) *Observation of M value by epochs*

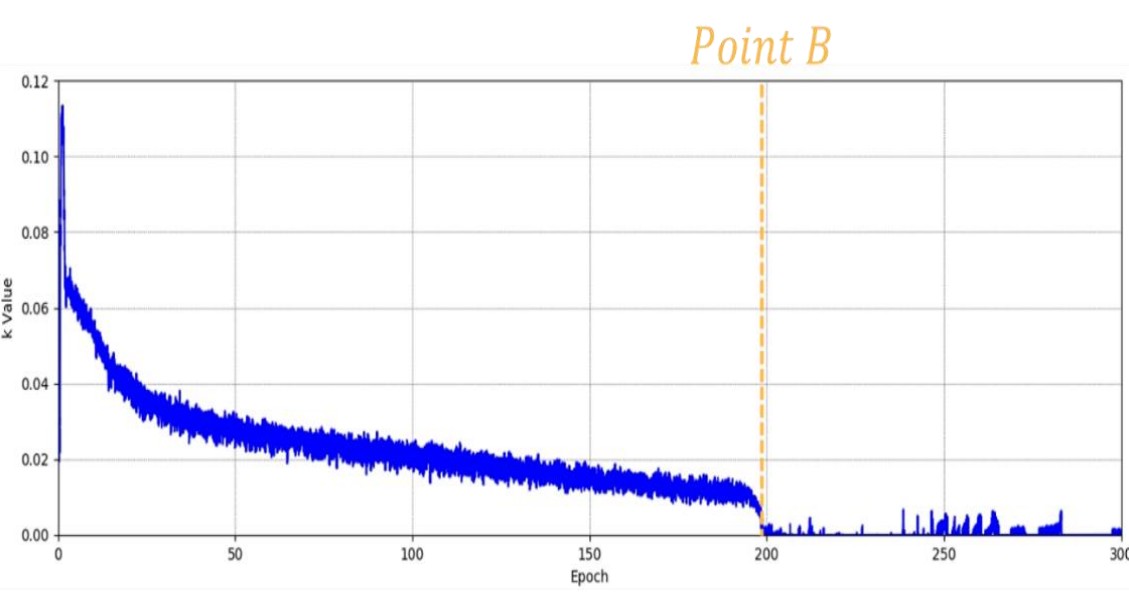

(b) *Observation of k value by epochs*

**Figure 16.** Results of convergence measurement for the BEGAN-CS (Boundary Equilibrium Generative Adversarial Network with Constrained Space) model.

At the $k$ value of Figure 16b, evidence of mode collapse was found. $k$ is a positive weight that determines the importance of $\mathcal{L}(G(z_D))$. As a result, mode collapse occurred exactly at 199 epochs Point B, where the $k$ value dropped suddenly.

Figure 17 shows the convergence measurement results of the proposed model. As with BEGAN-CS, we used L2 loss as reconstruction loss. The model proposed in (a) is most optimal when $M_{global}$ is 1.12, which is 0.38 lower than BEGAN-CS in Figure 16a. The $M_{global}$ value converges more as it approaches zero. Nonetheless, it is difficult to determine whether the model converges or collapses with $M_{global}$ values. In Figure 17b, there was also no section wherein the $k$ value suddenly dropped.

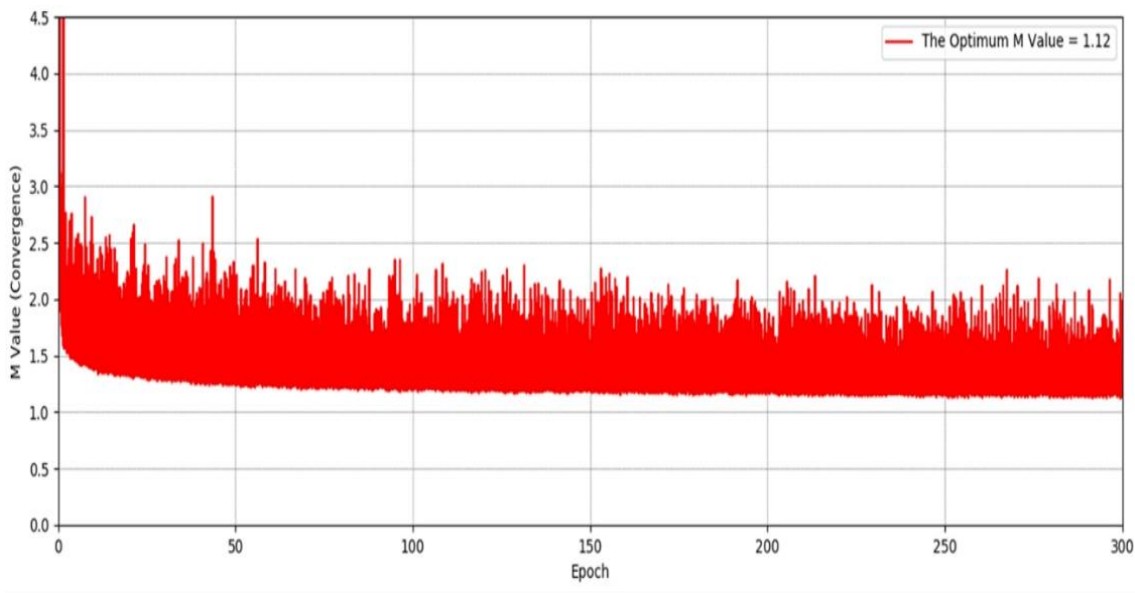

(a) Observation of M value by epochs

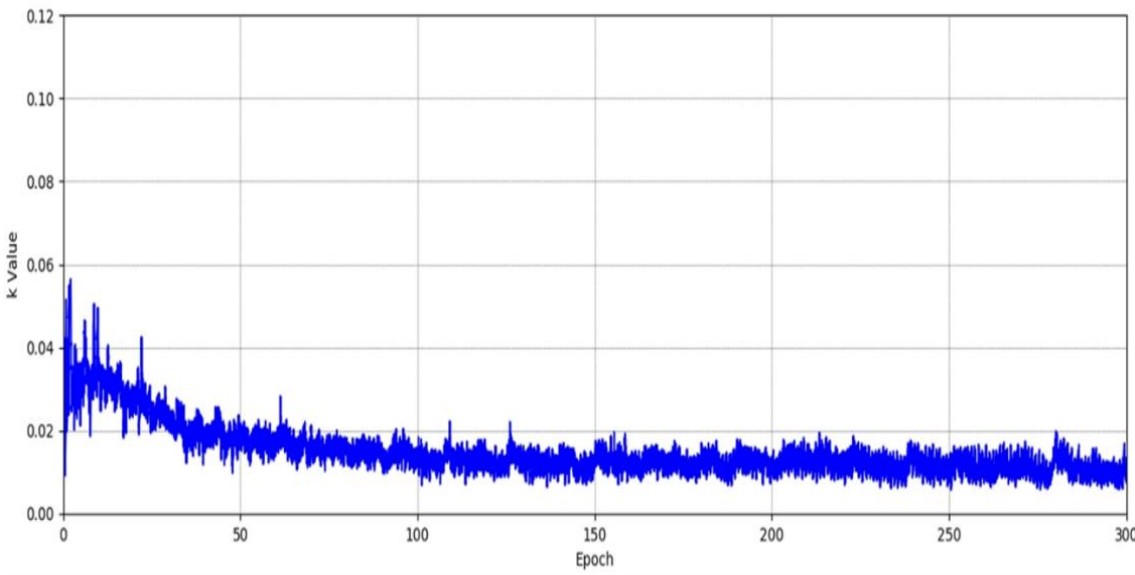

(b) Observation of k value by epochs

**Figure 17.** Convergence measurement results of the proposed model using L2 loss.

The smaller the fluctuation in value is, the higher the training stability. Both Figure 17a,b have a smaller range of fluctuations in value than BEGAN-CS. The proposed model had no mode collapse phenomenon at up to 300 epochs and converged better than BEGAN-CS.

Figure 18 presents the convergence measurement results of the proposed model. The model proposed in Figure 18a is most optimal when the $M_{global}$ value is 0.08. This is 1.42 lower than Figure 16a and 0.04 lower than Figure 17a. It is difficult to determine by the $M_{global}$ value whether the model in Figure 18 converges or collapses. In Figure 18b, there was also no section where the $k$ value suddenly dropped.

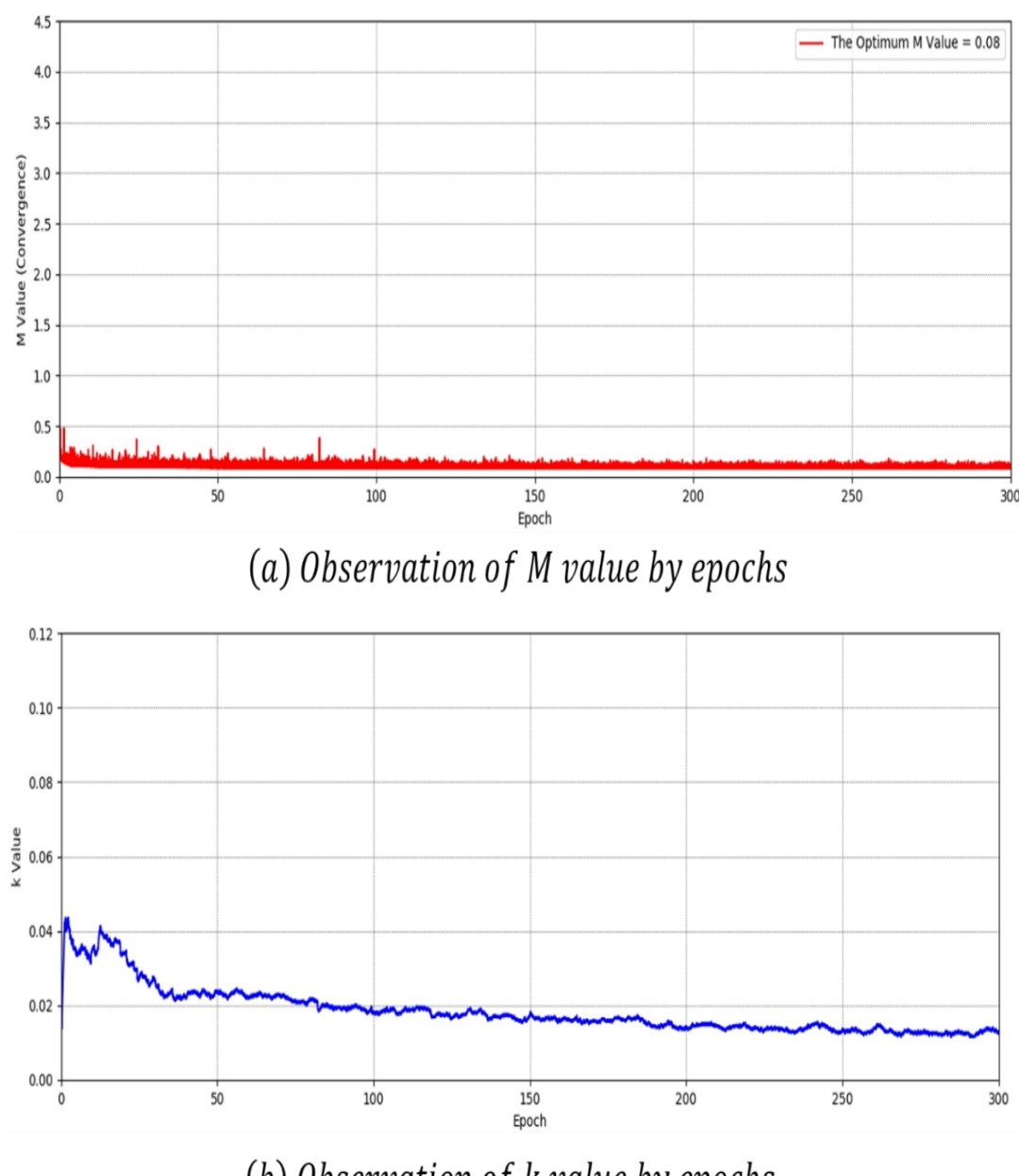

(a) *Observation of M value by epochs*

(b) *Observation of k value by epochs*

**Figure 18.** Convergence measurement results of the proposed model using L1 loss.

Even when L1 loss is applied to the proposed model, the mode collapse phenomenon does not occur up to 300 epochs. Moreover, (a) and (b) of Figure 18 converged in a better state with smaller fluctuations in values than (a) and (b) of Figures 16 and 17.

Figure 19 compares the generated images for specific times A, B, and C in the proposed model and BEGAN-CS. Previously, we used $M_{global}$ to confirm whether the model converged or collapsed, but we could not find any evidence of mode collapse from the convergence measurement value. The proposed model and BEGAN-CS saw no mode collapse, loss of diversity, or deterioration of quality up to 100 epochs.

At 199 epochs, however, mode collapse occurred for BEGAN-CS. If mode collapse occurs, optimization is not possible. In contrast, the proposed model saw no mode collapse up to 300 epochs. The proposed model showed a stable training process, and it was well versed in producing high-quality images. In addition, the proposed model showed convergence in a better state than BEGAN-CS. Table 3 compares the experimental results of the proposed model and BEGAN-CS.

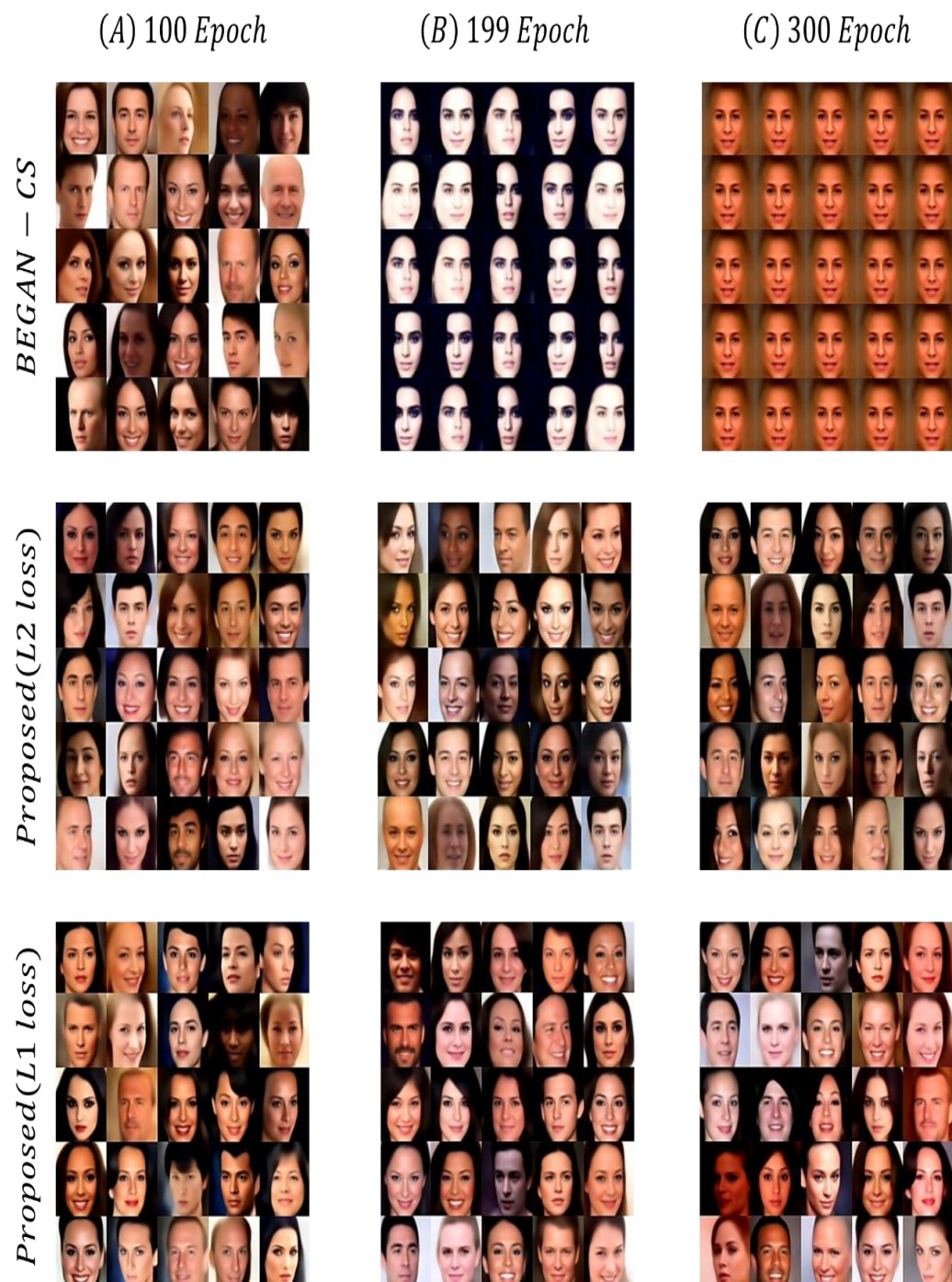

**Figure 19.** Observation results of mode collapse by epoch.

**Table 3.** Comparison of experimental results between the proposed model and BEGAN-CS.

| Model | Optimal M Value | Number of Parameters | Training Time | Mode Collapse Occurrence |
|---|---|---|---|---|
| **BEGAN-CS** | 1.50 | 20,629,320 | 9 days, 1 h, 45 min | 199 epochs |
| **Proposed (L2 loss)** | 1.12 | 10,331,144 | 8 days, 7 h, 05 min | - |
| **Proposed (L1 loss)** | 0.08 | 10,331,144 | 8 days, 6 h, 30 min | - |

There are three models in all, and training was done up to 300 epochs. We applied a model checkpoint algorithm to store the model for each epoch and set it to search for the optimal $M_{global}$ value automatically.

As a result of training, the $M_{global}$ value of the proposed model was 1.12 when L2 loss was used and 0.08 when L1 loss was used. The $M_{global}$ value of BEGAN-CS was 1.50, recording the lowest convergence accuracy. The proposed model shows a performance difference of 1.4 with BEGAN-CS at 0.38. The number of parameters was 10,331,144 for the proposed model, which was 10,298,176 fewer than the 20,629,320 for BEGAN-CS. Because of this, model capacity was also reduced by nearly half. The training of the proposed model took 8 days, 7 h, and 05 min when using L2 loss and 8 days, 6 h, and 30 min when using L1 loss. BEGAN-CS took the longest training time with 1 h and 45 min for 9 days. In contrast, training took only 18 h, 40 min and 19 h, 15 min with the proposed model. Finally, mode collapse occurred at 199 epochs for BEGAN-CS, whereas the proposed model saw no mode collapse up to 300 epochs.

## 6. Conclusions

The areas where GANs are utilized the most today are computer visions such as image style translation and image synthesis. Nowadays, they are also used to generate non-image data such as voice and natural language. BEGAN, which trains while balancing and regulating the generator and the discriminator, shows excellent performance in image synthesis. Even in BEGAN, however, there is a fundamental problem of GANs called mode collapse. The constraint loss of BEGAN-CS indicated that the mode collapse was solved, but the experimental results presented in this paper proved that such was not the case.

In this study, the BEGAN-CS discriminator structure was changed from AE to VAE in order to solve mode collapse. It also changed the structure of encoder and decoder, with the activation function changed from ELU to Leaky ReLU. The KLD term has been added to the loss function of the discriminator. In the KLD calculation process, hyperparameter *k* of BEGAN was added because it determines the importance of real and composite images.

The performance of the proposed model was 0.08 in convergent judging function $M_{global}$ when L1 losses were used; this was 1.42 lower than BEGAN-CS. In BEGAN-CS, the closer the $M_{global}$ value approaches zero, the higher it converges. In Figures 16–18, the $M_{global}$ value did not show evidence of convergence or collapsing. Mode collapse was identified by a variation graph of the *k* value. BEGAN-CS saw its *k* value drop sharply at 199 epochs, and mode collapse occurred. The proposed model had a constant *k* value for both L1 and L2 losses, and no mode collapse occurred. The proposed model was also more stable in training than BEGAN-CS, converging in a better state. Nonetheless, the experimental results in this paper are valid only for the CelebA dataset. This is because the experimental results may differ depending on the training dataset, and the optimal model structure and hyperparameters can also be changed.

The proposed model is also expected to achieve higher performance through additional training and hyperparameter adjustment. Future research needs to develop evaluation criteria that can identify large and small changes in color, texture, and pattern by supplementing the shortcomings of IS and FID. Furthermore, it is necessary to develop a model that can work well with various datasets. Applications including adding layers such as batch, layer, instance group normalization, dropout, and transpose convolution, quantifying objective functions and regular expressions, and changing the structure of discriminators like $\beta$ and total correlation-$\beta$VAE are expected to increase.

**Author Contributions:** Conceptualization, S.-W.P., J.-H.H. and J.-C.K.; Data curation, S.-W.P., J.-H.H. and J.-C.K.; Formal analysis, S.-W.P.; Investigation, S.-W.P.; Methodology, S.-W.P., J.-H.H. and J.-C.K.; Project administration, J.-H.H. and J.-C.K.; Resources, S.-W.P. and J.-C.K.; Software, S.-W.P., J.-H.H. and J.-C.K.; Supervision, J.-H.H. and J.-C.K.; Validation, J.-C.K.; Visualization, J.-H.H. and J.-C.K.; Writing–original draft, S.-W.P., J.-H.H. and J.-C.K.; Writing–review & editing, J.-H.H. and J.-C.K. All authors have read and agreed to the published version of the manuscript.

**Funding:** This work was supported by the National Research Foundation of Korea (NRF) grant funded by the Korea government (MSIT) (No.2017R1C1B5077157). Also, this research was supported by Energy Cloud R&D Program through the National Research Foundation of Korea (NRF) funded by the Ministry of Science, ICT (NRF-2019M3F2A1073385).

**Conflicts of Interest:** The authors declare no conflict of interest.

## Abbreviations

The following abbreviations are used in this manuscript:

| | |
|---|---|
| BEGAN-CS | Boundary Equilibrium Generative Adversarial Network with Constrained Space |
| AE | AutoEncoder |
| VAE | Variational AutoEncoder |
| IBM | International Business Machines |
| GANs | Generative Adversarial Networks |
| BEGAN | Boundary Equilibrium Generative Adversarial Networks |
| KLD | Kullback–Leibler Divergence |
| ELU | Exponential Linear Unit |
| MNIST | Modified National Institute of Standards and Technology |
| CIFAR | Canadian Institute For Advanced Research |
| CNN | Convolutional Neural Network |
| LAPGAN | Laplacian Pyramid of Generative Adversarial Network |
| DCGAN | Deep Convolutional Generative Adversarial Network |
| MLP | Multi-Layer Perceptron |
| cGAN | Conditional Generative Adversarial Nets |
| ReLU | Rectified Linear Unit |
| PReLU | Parametric Rectified Linear Unit |
| RReLU | Randomized leaky Rectified Linear Unit |
| Adam | Adaptive moment estimation |
| AdaGrad | Adaptive Gradient |
| CelebA | large-scale Celeb faces' Attributes |
| CUDA | Compute Unified Device Architecture |
| cuDNN | cuda Deep Neural Network library |
| EBGAN | Energy-Based Generative Adversarial Networks |
| CEGAN | Calibrating Energy-based Generative Adversarial Networks |
| AVB | Adversarial Variational Bayes |
| VEEGAN | Variational Encoder Enhancement to Generative Adversarial Networks |
| IS | Inception Score |
| FID | Frechet Inception Distance |

## Appendix A  More Generation

*Comparison of LSUN BEDROOM and CelebA Dataset*

Figures A1–A3 compares the results of image generation for each epoch in the LSUN bedroom and CelebA datasets. The LSUN bedroom dataset consisted of 162,770 training images, 19,867 validation images, and 19,962 test images, identical to the CelebA dataset. In addition, both datasets used L2 loss as reconstruction loss.

The mini batch size was set to 128, the epochs to 200, and the resolution of the training and prediction images to $64 \times 64$ pixels. The optimizer used Adam and set learning rate = "0.0001", $\beta 1$ = "0.9", $\beta 2$ = "0.999", and epsilon = $10^{-8}$. The bias was set to all 0. The weight initialization used Glorot's uniform distribution. The weights were adjusted to constant per layer as the training progressed. The learning rate of the generator and the discriminator is the same. The magnitude of input vector $z$ was set to 64, and $\gamma$ to 0.5. Weight $k_t$ was set to 0.0, and learning rate $\lambda_k$ of $k_t$ was set to 0.001. $\alpha$ was set to 0.1. The last layer of the encoder was kept at $8 \times 8$ size. $N_h$ and $N_z$ were set to 128 and 64, respectively,

in the experiment. The model used in the experiment has 10,331,144 parameters. Both datasets used L2 loss as reconstruction loss. Other hardware and software specifications are the same as before.

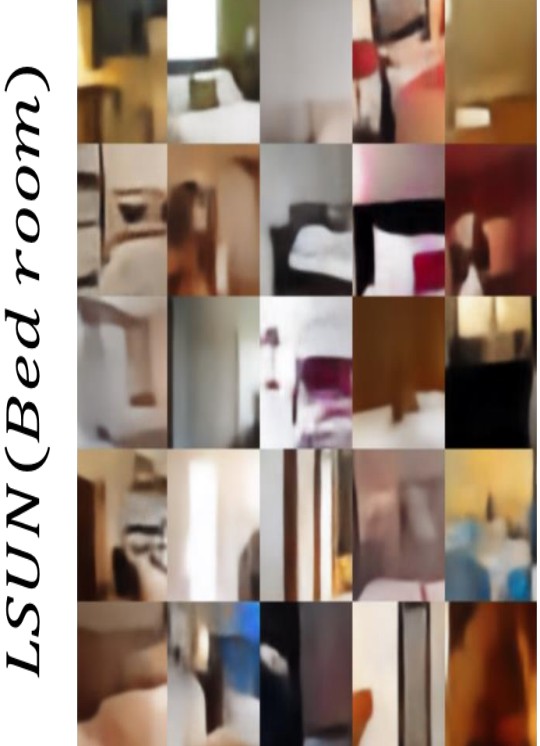
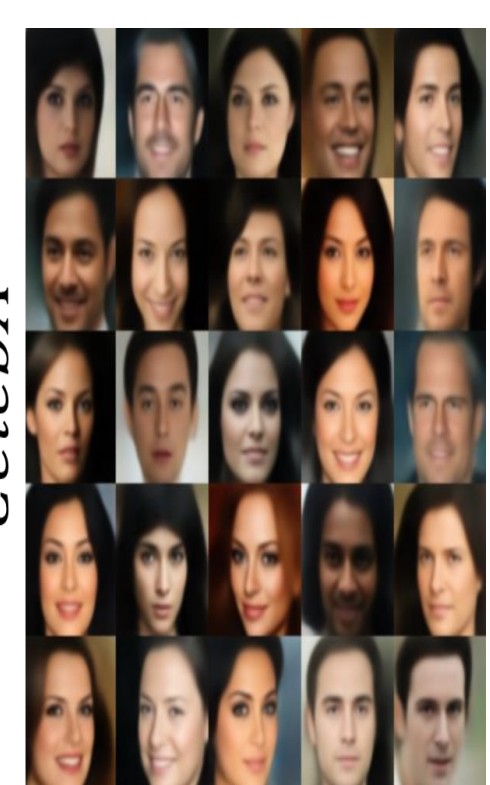

**Figure A1.** Image generation result of LSUN bedroom and CelebA datasets at 50 epoch.

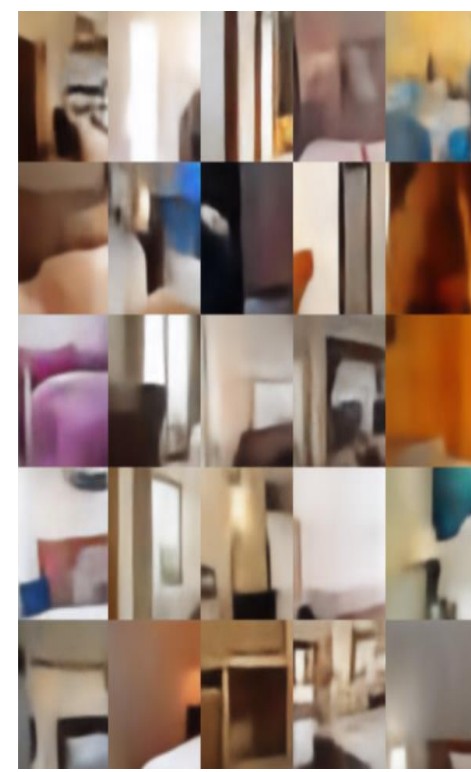
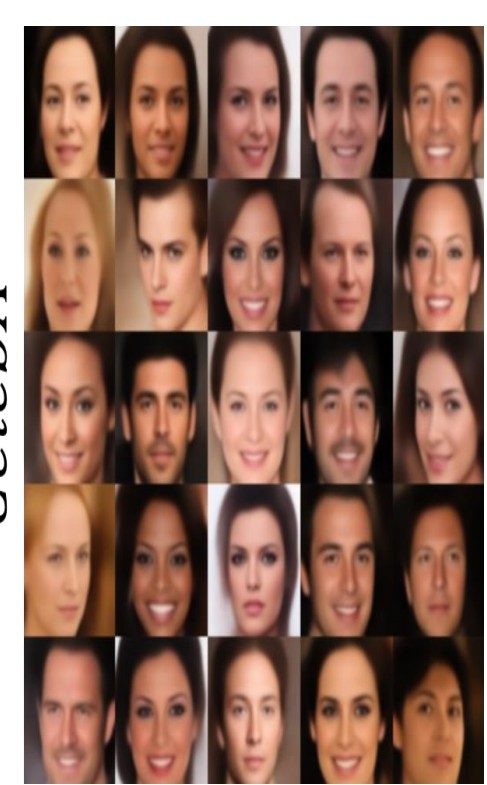

**Figure A2.** Image generation result of LSUN bedroom and CelebA datasets at 100 epoch.

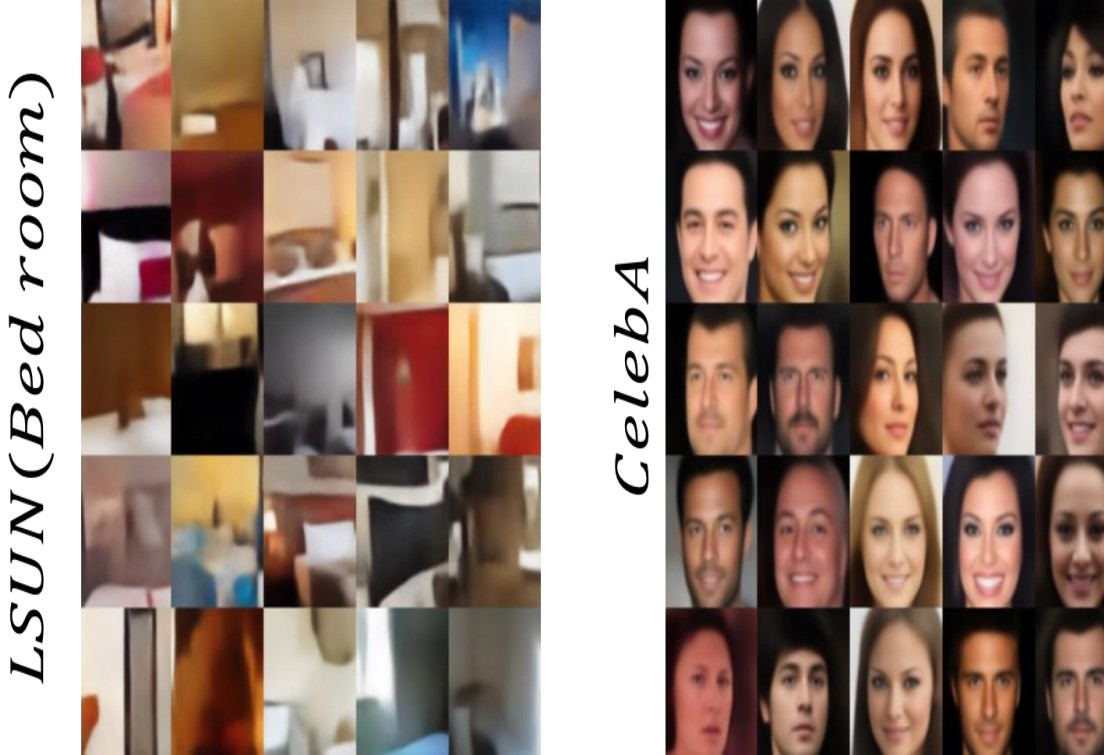

**Figure A3.** Image generation result of LSUN bedroom and CelebA datasets at 200 epoch.

Mode collapse did not occur in the LSUN bedroom image generated in the VAE up to 200 epoch. However, there was a 'blur' phenomenon that deteriorated the quality of the image. In Figures A1–A3, we determined that the experimental environment of the CelebA dataset is not suitable for generating the LSUN bedroom data.

Figures A4 and A5 present the convergence measurement results of LSUN bedroom and CelebA dataset. Similarly, it is difficult to determine whether both datasets converges or collapses with the $M_{global}$ value. In both datasets, there was no section in which the value of $k$ suddenly dropped in (b). In order to generate a high-level image, it is necessary to select the optimal data pre-processing, algorithm, and hyperparameter to train the model. The proposed model is also expected to achieve higher performance through additional training and hyperparameter adjustment. Future research needs to develop models that work well on a variety of datasets.

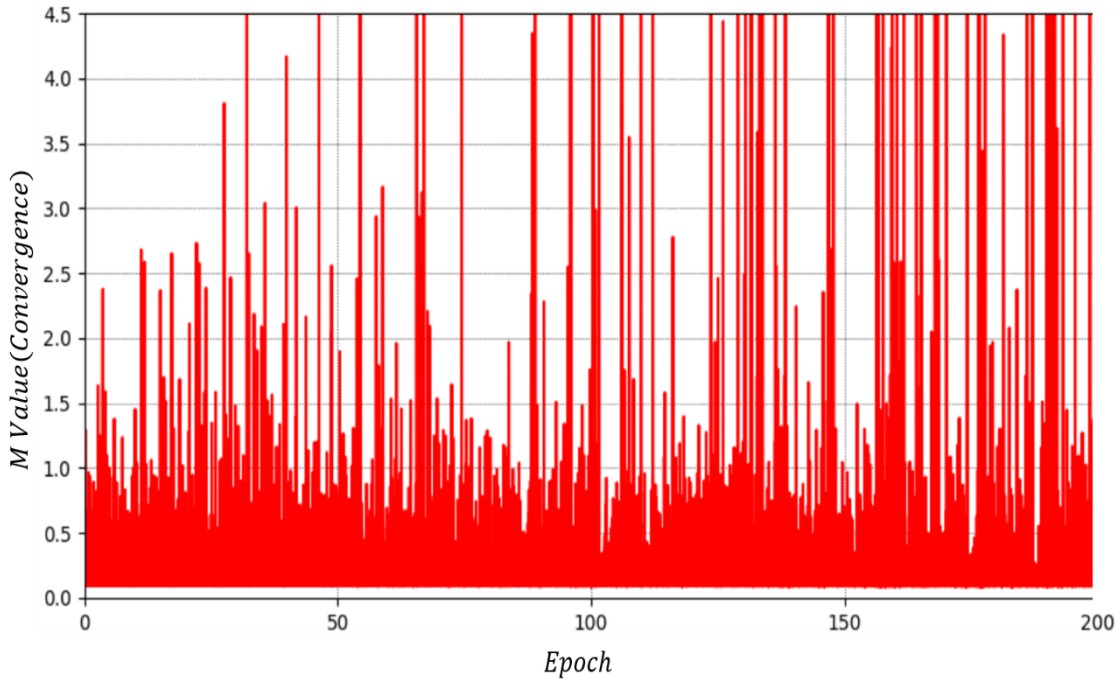

(*a*) *Observation of M value by epochs*

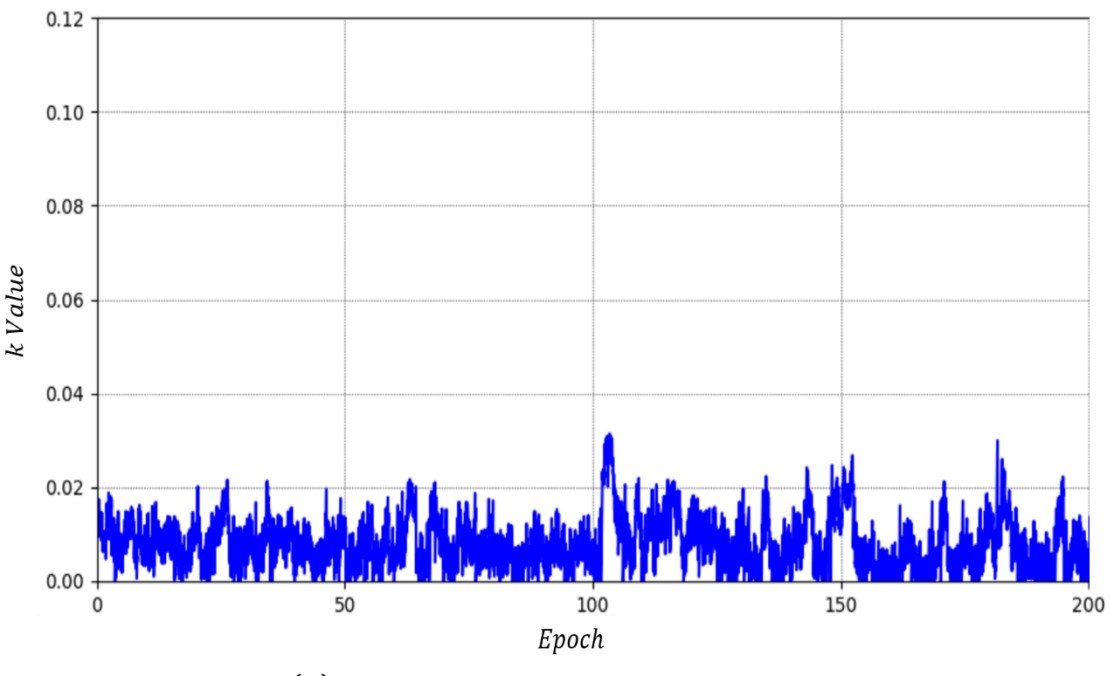

(*b*) *Observation of k value by epochs*

**Figure A4.** Results of convergence measurement for the LSUN bedroom dataset.

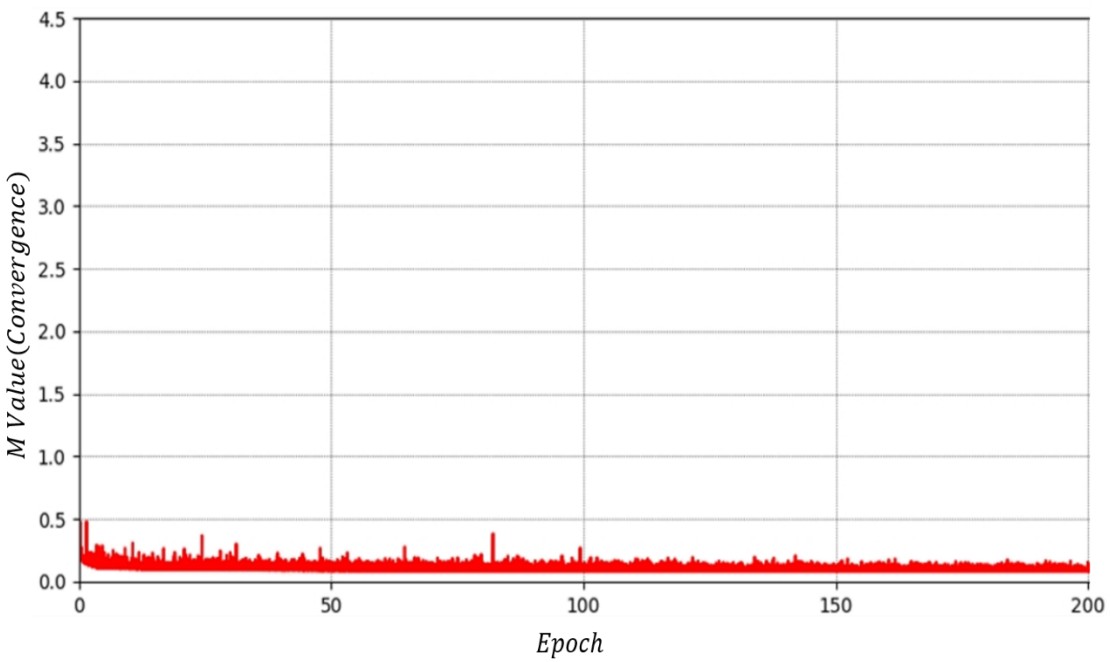

(a) *Observation of M value by epochs*

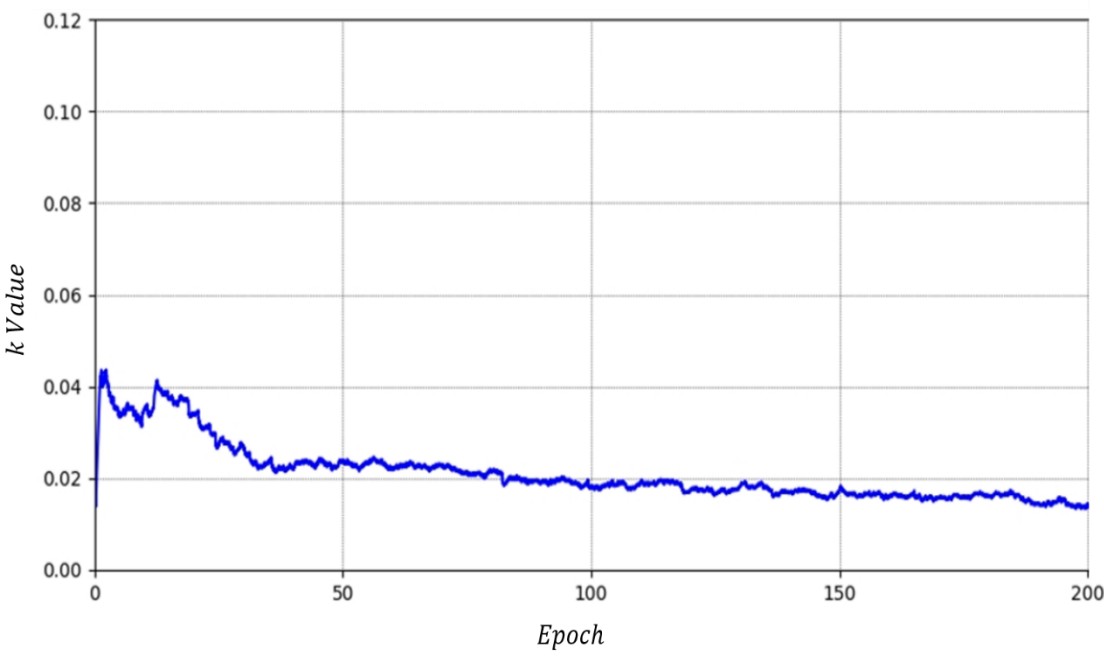

(b) *Observation of k value by epochs*

**Figure A5.** Results of convergence measurement for the CelebA dataset.

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
