# Peer review of "BEGAN v3: Avoiding Mode Collapse in GANs Using Variational Inference"

_electronics, doi:10.3390/electronics9040688_

Round 1

Reviewer 1 Report

Dear authors, thank you very much for the corrections made to the document. The topic covered in this paper is of great scientific interest, but the explanation and general development of the article remains unclear and difficult to understand in some paragraphs, even for someone with advanced knowledge of neural networks and deep learning.

Although some bibliographical reference is noted in the introduction, the explanation of the initial problem should be explained in more detail, since it is the starting point of the work. I believe that a more thorough description of the problem that has motivated the study is necessary.

The fact that the results obtained serve only for a specific dataset and and problem, reduces the translatability and scientific soundness of the work.

To be published, the article needs further experiments with different data sets and an extensive evaluation of the results obtained on these other problems.

Author Response

Comments and Suggestions for Authors

Dear authors, thank you very much for the corrections made to the document.

  1. The topic covered in this paper is of great scientific interest, but the explanation and general development of the article remains unclear and difficult to understand in some paragraphs, even for someone with advanced knowledge of neural networks and deep learning.

Reply->

First of all, we are sorry for the inconvenience. If you tell me a paragraph that is not clear and difficult to understand, I will submit it in a clear, easy-to-understand way.

The co corresponding author [1], corresponding author [2] are majoring in engineering theories for computer engineering. Although one of the drawbacks of the computer engineering-related studies is that the descriptions and explanations can be quite lengthy, but their advantage is that the contents of the proposal can be understood clearly just by reading them.

[1] Jun-Ho Huh

https://scholar.google.co.kr/citations?user=cr5wjNYAAAAJ&hl=ko

[2] Jong Chan Kim

https://scholar.google.co.kr/citations?user=YGPImkUAAAAJ&hl=ko

At the same time, to cover the drawbacks, the contribution parts have been included in every possible section while correcting the contents with the help of a Native English Speaker to improve readability within a limited time frame. It seems that the special edition ‘Electronic Solutions for Artificial Intelligence Healthcare [3]’, to which we’ve submitted our study, was to provide some understanding to the graphic engineers studying computer engineering from the energy point of view. Thus, we’ve included relevant discussions to make the paper more meaningful. The revised or added parts are being highlighted in red for your possible re-review.

  1. Although some bibliographical reference is noted in the introduction, the explanation of the initial problem should be explained in more detail, since it is the starting point of the work. I believe that a more thorough description of the problem that has motivated the study is necessary.

Reply->

Added a more detailed description of the initial problem that motivated the study. The contents added are as follows.

Add 1) Mode collapse is a phenomenon in which the generator generates only a few or a single image, and is divided into partial collapse and complete collapse. In general, if the generator is updated every step, the discriminator initially assigns a low probability to the previous output of the generator, so the generator appears as a cycle of convergence or endless mode hopping. When a mode collapse occurs, the discriminator penalizes the images generated in the mode to increase the loss of the generator, and the generator moves to another mode to avoid it. This is called mode hopping. Mode collapse has emerged as a fundamental problem for GANs. Therefore, in this paper, research was conducted to alleviate or solve mode collapse.

  1. The fact that the results obtained serve only for a specific data set and problem, reduces the translatability and scientific soundness of the work. To be published, the article needs further experiments with different data sets and an extensive evaluation of the results obtained on these other problems.

Reply->

Thank you for your considerate and deep insight. We’ve experienced difficulty as the image created after training a LSUN dataset with a BEGAN-CS model which was a target for comparison in the early stage of research was not realistic. Thus, the model introduced in this study was developed focusing on the facial dataset of CelebA in the process of solving the problem of ‘mode collapse’. Nonetheless, to aovid a hasty generalization, the following statements have been added to the conclusion part while correcting the manuscript with the help of a native English speaker to enhance readabilty. We’d appreciate very much if you will review our paper again in this regard.

The sataements added are as follows:

  1. “However, the obtained results are valid only for the dataset of CelebA.”
  2. “The experimental results can be varied depending on the training datasets as the optimal model structure and hyper parameter could be changed.”  
  3. “Further, development of a model that can deal with a variety of datasets is necessary. “ was added as a future research task to be conducted.

Also, We add)

Add 1) The characteristics of deep learning models can be divided into discriminative and generative models for comparison. discriminative models distinguish and classify differences in input patterns. If enter an image of a dog, determine the dog with a specific probability. The training model is trained to maximize the probability that a label called will be output when given the data . In other words, the discriminative model is a deep learning model that classifies or recognizes data and directly modeling conditional probability . Generative models, on the other hand, contain more information than discriminative models. Knowing the distribution of joint allows you to find the conditional probability and the distribution of the data itself. The generative model can understand and explain the structure of the input data.

Add 2) Mode collapse is a phenomenon in which the generator generates only a few or a single image, and is divided into partial collapse and complete collapse. In general, if the generator is updated every step, the discriminator initially assigns a low probability to the previous output of the generator, so the generator appears as a cycle of convergence or endless mode hopping. When a mode collapse occurs, the discriminator penalizes the images generated in the mode to increase the loss of the generator, and the generator moves to another mode to avoid it. This is called mode hopping. Mode collapse has emerged as a fundamental problem for GANs. Therefore, in this paper, research was conducted to alleviate or solve mode collapse.

Reviewer 2 Report

I have seen a significant improvement in this revision, and this time they have seriously addressed my concerns. I would then like to accept this paper.

Author Response

Comments and Suggestions for Authors

I have seen a significant improvement in this revision, and this time they have seriously addressed my concerns. I would then like to accept this paper.

Reply->

First of all, we are sorry for the inconvenience. We reviewed whether the introduction provided sufficient background and if there were any missing references. As a result of the review, we locate that the description of the characteristic of the deep learning model is missing. Therefore, the characteristic of two deep learning models (generative model, discriminative model) were added as follows. Also, added a more detailed description of the initial problem that motivated the study.

Add 1) The characteristics of deep learning models can be divided into discriminative and generative models for comparison. discriminative models distinguish and classify differences in input patterns. If enter an image of a dog, determine the dog with a specific probability. The training model is trained to maximize the probability that a label called will be output when given the data . In other words, the discriminative model is a deep learning model that classifies or recognizes data and directly modeling conditional probability . Generative models, on the other hand, contain more information than discriminative models. Knowing the distribution of joint allows you to find the conditional probability and the distribution of the data itself. The generative model can understand and explain the structure of the input data.

Add 2) Mode collapse is a phenomenon in which the generator generates only a few or a single image, and is divided into partial collapse and complete collapse. In general, if the generator is updated every step, the discriminator initially assigns a low probability to the previous output of the generator, so the generator appears as a cycle of convergence or endless mode hopping. When a mode collapse occurs, the discriminator penalizes the images generated in the mode to increase the loss of the generator, and the generator moves to another mode to avoid it. This is called mode hopping. Mode collapse has emerged as a fundamental problem for GANs. Therefore, in this paper, research was conducted to alleviate or solve mode collapse.

This manuscript is a resubmission of an earlier submission. The following is a list of the peer review reports and author responses from that submission.

Round 1

Reviewer 1 Report

In this article the authors propose several solutions to avoid the mode collapse problem common of Boundary Equilibrium Generative Adversarial Networks with Constrained Space (BEGAN-CS). More in detail, they:

1 - replace the discriminator structure from an auto-encoder (AE) to a variational auto-encoder (VAE);

2 - change the activation function of encoder and decoder from ELU to Leaky RELU;

3 - add a Kullback-Leibler Divergence (KLD) term to the discriminator loss

The achieved results show the effectiveness of the method, however the solution is not very innovative and the document does not read well. A substantial rereading must be done. First of all, I strongly invite the authors to just not copy-paste the text from the abstract to the introduction or vice-versa because this is redundant and it does not give any new information.

Even if in the introduction is explained what a BEGAN is, the full name for BEGAN-CS is not given until section 2 name. I kindly suggest the authors to include the full name before the acronym is used.

In line 136, a reference to the "theory of proportional control" should be added.

This sentence "?? is the learning rate about ?." in line 120 is not clear. Is it appropriate to replace "about" with "for"?

Variational inference reference should be added in line 193.

The resolution of input image x is not given in Figure 1.

In line 268, the authors should explain the sentence "The convolution operation was used from 1 to 4".

In line 291, the following sentence is not clear: "The training took 7 hours and 05 minutes for 8 days and 6 hours and 30 minutes for 8 days, respectively."

In line 312, the authors claim that the resolution of training and predicted images is 64 x 64 pixels, then it is not clear why in Figure 8 and 9, showed images are rectangles.

Extensive results should be reported also for another task such as the LSUN-Bedrooms dataset.

Check again all the references in particular:

  • the organization is missed in several conference refs, such as [4,7,14,20]
  • Publisher is missed in journal article ref 32

Author Response

Reviewer 1) English language and style

(x) Extensive editing of English language and style required

Comments and Suggestions for Authors

The achieved results show the effectiveness of the method, however the solution is not very innovative and the document does not read well.

-----

Reply-----

Dear Reviewer

First of all, thank you for reviewing our paper in detail and giving us an appropriate comment. The co corresponding author [1], corresponding author [2] are majoring in engineering theories for computer engineering. Although one of the drawbacks of the computer engineering-related studies is that the descriptions and explanations can be quite lengthy, but their advantage is that the contents of the proposal can be understood clearly just by reading them.

[1] Jun-Ho Huh

https://scholar.google.co.kr/citations?user=cr5wjNYAAAAJ&hl=ko

[2] Jong Chan Kim

https://scholar.google.co.kr/citations?user=YGPImkUAAAAJ&hl=ko

At the same time, to cover the drawbacks, the contribution parts have been included in every possible section while correcting the contents with the help of a native English speaker to improve readability within a limited time frame. It seems that the special edition ‘Electronic Solutions for Artificial Intelligence Healthcare [3]’, to which we’ve submitted our study, was to provide some understanding to the graphic engineers studying computer engineering from the energy point of view. Thus, we’ve included relevant discussions to make the paper more meaningful. The revised or added parts are being highlighted in red for your possible re-review.

Also, I am really sorry that it's not an innovative method and it's not easy to read a paper. However, thank you for spending a lot of time in the review. All the contents that the judge pointed out were considered correct, and they were all corrected.

Could you read again?

First of all, I strongly invite the authors to just not copy-paste the text from the abstract to the introduction or vice-versa because this is redundant and it does not give any new information.

-----

Reply -----

We are very sorry that the judge are uncomfortable reading the paper. The abstract has been completely revised to reflect the judge points. The modified contents are as follows.

Add) In the field of deep learning, the generative model did not attract much attention until GANs appeared. In 2014, Google's Ian Goodfellow proposed a generative model called GANs. GANs uses a different structure and objective function than the existing generative model. For example, GANs uses two neural networks: a generator that creates a realistic image and a discriminator that distinguishes whether the input is real or synthetic. If there are no problems in the training process, GANs can generate images that are difficult for even experts to distinguish between authenticity. Currently, GANs is the most researched in the field of computer vision, which deals with the technology of image style translation, synthesis and generation, and various models have been announced. The issues raised are also improving one by one. In image synthesis, BEGAN(Boundary Equilibrium Generative Adversarial Networks), which outperforms previously announced GANs, learns the latent space of the image while balancing the generator and discriminator. However, BEGAN also has a mode collapse in which the generator generates only a few or a single image. Although BEGAN-CS(Boundary Equilibrium Generative Adversarial Network with Constrained Space), which was improved in terms of loss function, was introduced, it did not solve the mode collapse. The discriminator structure of BEGAN-CS is AE, which cannot create a particularly useful or structured latent space. Compression performance is not good either. In this paper, it is considered that this characteristic of AE is related to the occurrence of mode collapse. Thus, we used VAE (Variational AutoEncoder), which added statistical techniques to AE. As a result of the experiment, the proposed model did not cause mode collapse, and converged to a better state than BEGAN-CS.]

Even if in the introduction is explained what a BEGAN is, the full name for BEGAN-CS is not given until section 2 name. I kindly suggest the authors to include the full name before the acronym is used.

-----

Reply ----- The full name has been modified to appear first in abstract and introduction. I'm sorry.

Before : Although BEGAN-CS

After : Although BEGAN-CS(Boundary Equilibrium Generative Adversarial Network with Constrained Space)

In line 136, a reference to the "theory of proportional control" should be added.

-----

Reply ----- The following explanation of the theory of A was added to the paper.

Add 1) “The proportional control theory is one of the automatic control methods, and the more the force deviates from the target point, the greater the force to return to the target point.”

It is a good idea to summarize the terms on page 4 in a table neatly (and also denoting it in This sentence "?? is the learning rate about ?." in line 120 is not clear. Is it appropriate to replace "about" with "for"?

-----

Reply ----- I am very sorry for my poor English writing skills. The following changes have been made to reflect the feedback of the judge.

Before :  is the learning rate about .

After :  is the learning rate for .

Variational inference reference should be added in line 193.

-----

Reply ----- Thank you for a good point. Looking back, we think the point you pointed out is correct. References to variance inference have been added to line 193.

Before : The variational inference is a problem of finding an estimation distribution  that is close to the posterior distribution and can be expressed as in Equation (14).

After : The variational inference[11] is a problem of finding an estimation distribution  that is close to the posterior distribution and can be expressed as in Equation (14).

The resolution of input image x is not given in Figure 1.

-----

Reply ----- The resolution of the input image has been added to Figure 1. We made a mistake not to display the resolution of the input video. I'm really sorry.

Figure 1. Discriminator structure of the proposed model

In line 268, the authors should explain the sentence "The convolution operation was used from 1 to 4".

-----

Reply ----- The following sentence was added to the paper.

Add 1) “'1' means 1x1 convolution, and '4' means 4x4 convolution”

In line 291, the following sentence is not clear: "The training took 7 hours and 05 minutes for 8 days and 6 hours and 30 minutes for 8 days, respectively."

-----

Reply ----- Sorry for omitting the subject. We revised the sentence by adding the subject. Thank you.

Before : The training took 7 hours and 05 minutes for 8 days and 6 hours and 30 minutes for 8 days, respectively.

After : Training time was 8 days 7 hours 5 minutes when L2 loss was used, and 8 days 6 hours 30 minutes when L1 loss was used.

In line 312, the authors claim that the resolution of training and predicted images is 64 x 64 pixels, then it is not clear why in Figure 8 and 9, showed images are rectangles.

-----

Reply ----- “Figures 8-9 show the result of outputting 64 images of 64x64 pixels at a time.” We added this sentence to paper. Thank you. Capture part of the code and attach it.

Extensive results should be reported also for another task such as the LSUN-Bedrooms dataset.

-----

Reply ----- Thank you for the judge high insight and sincere and sharp criticism. As a result of learning the LSUN data set with the BEGAN-CS model to be compared in the initial study, it was difficult to produce a realistic image. So, the model of this paper was developed by focusing on CelebA, the human face data set, in the process of solving mode collapse. However, in order not to make a hasty generalization error, the following sentence was added to the conclusion.

Add 1) However, the experimental results in this paper are valid only for the CelebA data set.

Add 2) This is because the experimental results may differ depending on the training data set, and the optimal model structure and hyperparameters can also be changed.

Add 3) Furthermore, it is necessary to develop a model that can work well with various datasets.

the organization is missed in several conference refs, such as [4,7,14,20]

-----

Reply -----

[4] → [J. Shen, R. Pang, R. J. Weiss, M. Schuster, N. Jaitly, Z. Yang, et al., "Natural TTS Synthesis by Conditioning WaveNet on Mel Spectrogram Predictions," arXiv preprint, arXiv:1712.05884v2, 2018.]

[7] → [C.-C. Chang, C. H. Lin, C.-R. Lee, D.-C. Juan, W. Wei, and H.-T Chen, "Escaping from Collapsing Modes in a Constrained Space," arXiv preprint, arXiv:1808.07258, 2018.]

[14] → [K. He, X. Zhang, S. Ren, and J. Sun, “Delving Deep Into Rectifiers: Surpassing Human-Level Performance On Imagenet Classification,” arXiv preprint, arXiv:1502.01852v1, 2015.]

[20] → [Z. Liu, P. Luo, X. Wang, and X. Tang, "Deep Learning Face Attributes in the Wild," arXiv preprint, arXiv:1411.7766v3, 2015.]

Reviewer 2 Report

This paper inroduces a methodololgy to avoid Model Collapse Deep Learning systems with Generative Adversarial Netoworks. 

Although the problem is well presented and explained, it is necessary to add in the introduction what "mode collapse" is, since it is mentioned and presented as the problem to be solved in the article, but there is no explanation about what is and why it is a problem.

Chapter 2 begins very abruptly. It would be advisable to add text by entering the topic of the chapter.

The experiments have been performed with a very specific dataset (CelebA). Although the results obtained are good, in the conclusions should not speak in a general way, since you should do tests with more datasets for this.

Author Response

Comments and Suggestions for Authors

Although the problem is well presented and explained, it is necessary to add in the introduction what "mode collapse" is, since it is mentioned and presented as the problem to be solved in the article, but there is no explanation about what is and why it is a problem.

-----

Reply -----

-----

Dear Reviewer

First of all, thank you for reviewing our paper in detail and giving us an appropriate comment. The co corresponding author [1], corresponding author [2] are majoring in engineering theories for computer engineering. Although one of the drawbacks of the computer engineering-related studies is that the descriptions and explanations can be quite lengthy, but their advantage is that the contents of the proposal can be understood clearly just by reading them.

[1] Jun-Ho Huh

https://scholar.google.co.kr/citations?user=cr5wjNYAAAAJ&hl=ko

[2] Jong Chan Kim

https://scholar.google.co.kr/citations?user=YGPImkUAAAAJ&hl=ko

At the same time, to cover the drawbacks, the contribution parts have been included in every possible section while correcting the contents with the help of a native English speaker to improve readability within a limited time frame. It seems that the special edition ‘Electronic Solutions for Artificial Intelligence Healthcare [3]’, to which we’ve submitted our study, was to provide some understanding to the graphic engineers studying computer engineering from the energy point of view. Thus, we’ve included relevant discussions to make the paper more meaningful. The revised or added parts are being highlighted in red for your possible re-review.

Also, in the introduction, we added what is the mode collaps and why it is a problem when it occurs.

Add 1) Mode collapse is a phenomenon in which the generator generates only a few or a single image.

Add 2) It is difficult to optimize a model that has a mode collapse once.

Add 3) This is because when the mode collapse occurs, the discriminator increases the loss of the generator, and the generated images in the mode are panelized and the generator moves to another mode to avoid this.

Chapter 2 begins very abruptly. It would be advisable to add text by entering the topic of the chapter.

-----

Reply -----

-----

We are very sorry that the developments were sudden and inconvenient to read. In order to make the start more natural, we added sentences related to the subject according to the high opinion of the judges. The measures are as follows.

Add 1) Both BEGAN and BEGAN-CS have not solved the mode collapse, but the proposed method is based on boundary equilibrium and constrained space algorithms.

Add 2) The characteristics of the proposed model are covered in the next chapter, and this chapter describes the structure and learning algorithm of BEGAN and BEGAN-CS.

The experiments have been performed with a very specific dataset (CelebA). Although the results obtained are good, in the conclusions should not speak in a general way, since you should do tests with more datasets for this.

-----

Reply -----

-----

Thank you for the correct point. The judge comments were treated as follows:

Add 1) However, the experimental results in this paper are valid only for the CelebA data set.

Add 2) This is because the experimental results may differ depending on the training data set, and the optimal model structure and hyperparameters can also be changed.

Add 3) Furthermore, it is necessary to develop a model that can work well with various datasets.

Reviewer 3 Report

The authors have tried to present an advance in GAN to deal with the model collapse. However, it is unacceptable in current shape. The presentation of the method is definitively unclear to the readers, and the structure of this paper is chaotic. Many equations and variables are used without clearly explaining their notation and relation among each other. Some examples are given in the following:

  1. In line 77, it is mentioned to replace ELU with Leaky ReLU, but the authors have not mentioned any motivation for this replacement.
  2. The sentence from line 96 to 98 is hardly to understand. 
  3. The "~" in Equation (2) is not defined.
  4. Do the "Wasserstein" and "Wasserstein distance" mean the same thing ? It is completely unclear for me. The notation "W" is firstly presented in Equation (2), and then explained with different names in line 103 and 106, respectively. It is the same case for the notations mu_1 and mu_2. What does the distribution mean ? Do they mean probability density function ?
  5. What are the z_D, z_G and z ? How can understand the sentence "z_D, z_G is the output from z." ? Is "z" a function ? 

This paper should fundamentally revised. Firstly, the authors have to present their signal model in a reasonable notation system. Then, in this framework the disadvantage in classical GAN can be explained, and the improvement can be accordingly proposed. 

Author Response

Comments and Suggestions for Authors.

In line 77, it is mentioned to replace ELU with Leaky ReLU, but the authors have not mentioned any motivation for this replacement.

-----

Reply -----

-----

Dear Reviewer

First of all, thank you for reviewing our paper in detail and giving us an appropriate comment. The co corresponding author [1], corresponding author [2] are majoring in engineering theories for computer engineering. Although one of the drawbacks of the computer engineering-related studies is that the descriptions and explanations can be quite lengthy, but their advantage is that the contents of the proposal can be understood clearly just by reading them.

[1] Jun-Ho Huh

https://scholar.google.co.kr/citations?user=cr5wjNYAAAAJ&hl=ko

[2] Jong Chan Kim

https://scholar.google.co.kr/citations?user=YGPImkUAAAAJ&hl=ko

At the same time, to cover the drawbacks, the contribution parts have been included in every possible section while correcting the contents with the help of a native English speaker to improve readability within a limited time frame. It seems that the special edition ‘Electronic Solutions for Artificial Intelligence Healthcare [3]’, to which we’ve submitted our study, was to provide some understanding to the graphic engineers studying computer engineering from the energy point of view. Thus, we’ve included relevant discussions to make the paper more meaningful. The revised or added parts are being highlighted in red for your possible re-review.

Also, it is thought that what you pointed out is correct enough. So we added the following sentence.

Add 1) "Leaky ReLU, the negative part of which is unsaturated, is thought to work better than ELU, the negative part of which is saturated, so I changed it."

The sentence from line 96 to 98 is hardly to understand.

-----

Reply -----

-----

Added a brief definition of Wasserstein distance as a sentence. Wasserstein distance.

Add 1) Wasserstein distance is the minimum cost of moving A's probability distribution to B's probability distribution.

The "~" in Equation (2) is not defined.

-----

Reply -----

-----

In relation to equation (2), We added the following sentence.

Add 1)

Add 2) ~ is a tilde, a symbol that describes the relationship between variables and distribution. Distribution  of  and distribution  of  is used to explain that it is drawn from .

Add 3) inf is short for Infimum, the greatest lower bound. That is, it is the largest value of the lower limit.

Add 4)  is the value expected as the average of the values that can be obtained by infinitely repeating a random process.

Do the "Wasserstein" and "Wasserstein distance" mean the same thing ? It is completely unclear for me. The notation "W" is firstly presented in Equation (2), and then explained with different names in line 103 and 106, respectively. It is the same case for the notations mu_1 and mu_2. What does the distribution mean ? Do they mean probability density function ?

-----

Reply -----

-----

Sorry for the great confusion. In the paper, the Wasserstein and Wasserstein distance are the same, so they were all unified as the Wasserstein distance. and have moved the description of the symbol to the previous sentence to avoid confusion. Added the sentence "distribution is a probability density function" to the paragraph where the distribution first appears.

What are the z_D, z_G and z ? How can understand the sentence "z_D, z_G is the output from z." ? Is "z" a function ?

-----

Reply -----

-----

I'm sorry I didn't write it correctly. In line 132, we added the following sentence.

Add 1)  is the output from .  is the noise vector that is the input of the generator or discriminator.

This paper should fundamentally revised. Firstly, the authors have to present their signal model in a reasonable notation system. Then, in this framework the disadvantage in classical GAN can be explained, and the improvement can be accordingly proposed.

-----

Reply-----

-----

The judge comments related to inappropriate markings were all reflected in the paper. Thank you. Regarding the disadvantages of traditional GANs, I added the following sentence.

Add 1) In BEGAN, there is a fundamental problem with GANs called mode collapse.

Add 2) This paper presents an alternative research on the fundamental problem of GANs.

Add 3) However, even in BEGAN, there is a fundamental problem of GANs called mode collapse.

Round 2

Reviewer 3 Report

After having gone through the authors' response and the revised paper, I am not satisfied with the current version. My questions are not clearly answered, and the paper still contains many findings in the way of presentation as well as its notation. I would not accept it in its current form.